# The residence time of water in the atmosphere revisited

Ruud J. van der Ent[1] and Obbe A. Tuinenburg[2]

[1]Department of Physical Geography, Faculty of Geosciences, Utrecht University, Utrecht, the Netherlands
[2]Department of Environmental Sciences, Copernicus Institute for Sustainable development, Utrecht University, Utrecht, the Netherlands

*Correspondence to:* Ruud J. van der Ent (r.j.vanderent@uu.nl)

**Abstract.** This paper revisits the knowledge on the residence time of water in the atmosphere. Based on state-of-the-art data of the hydrological cycle we derive a global average residence time of $8.9\pm0.4\,\mathrm{days}$ (uncertainty given as one standard deviation). We use two different atmospheric moisture tracking models (WAM-2layers and 3D-Trajectories) to obtain atmospheric residence time characteristics in time and space. The tracking models estimate the global average residence time to be around 8.5 days based on ERA-Interim data. We conclude that the statement of a recent study that the global average residence time of water in the atmosphere is 4–5 days, is not correct. We derive spatial maps of residence time, attributed to evaporation and precipitation, and age of atmospheric water, showing that there are different ways of looking at temporal characteristics of atmospheric water. Longer evaporation residence times often indicate larger distances towards areas of high precipitation. From our analysis we find that the residence time over the ocean is about 2 days lower than over land. It can be seen that in winter, the age of atmospheric moisture tends to be much lower than in summer. On the Northern Hemisphere, due to the contrast in ocean-to-land temperature and associated evaporation rates, the age of atmospheric moisture increases following atmospheric moisture flow inland in winter, and decreases in summer. Looking at the probability density functions of atmospheric residence time for precipitation and evaporation we find long-tailed distributions with the median around 5 days. Overall, our research confirms the 8–10 days traditional estimate for the global mean residence time of atmospheric water, and our research contributes to a more complete view on the characteristics of the turnover of water in the atmosphere in time and space.

## 1 Introduction

The time it takes before evaporated water from land and oceans is returned to the land surface as precipitation is a fundamental characteristic of the Earth's hydrological cycle. This atmospheric residence time of moisture is not often discussed in the scientific research literature. The global average residence time of atmospheric moisture is mostly seen as non-controversial knowledge in textbooks (e.g., Chow et al., 1988; Hendriks, 2010; Jones, 1997; Ward and Robinson, 2000), general water literature (e.g., Bodnar et al., 2013; Savenije, 2000) and educational web pages (e.g., UCAR, 2011). All of these examples estimate the global average residence time of atmospheric moisture based on the size of the atmospheric reservoir divided by the incoming or outgoing flux and as such arrive at estimates in the range of 0.022–0.027 years or 8–10 days.

While the global average residence time is a simple estimate, spatial and temporal pictures of residence times are more difficult to provide. Local depletion times, given by $W/P$, and local restoration times, given by $W/E$ (where $W$ is water in

the local atmospheric column, $P$ is precipitation and $E$ is evaporation), were computed near-globally by Trenberth (1998) and van der Ent and Savenije (2011). When horizontal moisture transport is small compared to precipitation and evaporation they provide a proxy for the residence time. However, it is safer to interpret them as local time scales of atmospheric moisture recycling (van der Ent and Savenije, 2011). Trenberth (1998) found a global average residence time of atmospheric moisture

of 8.9 days based on evaporation and 9.1 days based on precipitation, and attributed this difference to the input data having a non-closure of the global water balance. On the other hand, global spatial average local depletion times and restoration times were found to be 8.1 days and 8.5 days respectively. The difference with the global moisture weighted values was explained by the heterogeneous distribution of evaporation and precipitation over the Earth and their spatial correlation with atmospheric moisture.

Moisture tracking models also allow for the estimation of atmospheric moisture residence time. A semi-Lagrangian method of passive water vapor tracers in a general circulation model was used to perform a special experiment in which all atmospheric water was tagged, after which it was evaluated how fast the tagged water rains out (Bosilovich and Schubert, 2002; Bosilovich et al., 2002). To clarify, in this experiment the atmosphere was replenished by evaporation, but this water was not tagged. The outcome of this analysis was a global average residence time of 8.5 days in May (Bosilovich and Schubert, 2002) and

9.2 days in an undefined time period (Bosilovich et al., 2002) respectively. The same types of experiments were performed by Yoshimura et al. (2004) with an offline 1-layer Eulerian moisture tracking model using the global GAME reanalysis data. They found a global mean residence time between 7.3 days (April) and 9.2 days (August).

The global experiments of Bosilovich et al. (2002) and Yoshimura et al. (2004) also suggest that the time it takes before atmospheric water to rain out has a negative exponential distribution, which belongs to a Poisson process (e.g., Goessling

and Reick, 2013a). The median residence time can be estimated from the graphs provided by Bosilovich et al. (2002) and Yoshimura et al. (2004) as around 6 days. On a more local scale with a pure Lagrangian moisture tracking method applied in India, Tuinenburg et al. (2012) showed probability density functions of the time that evaporated water remains in the atmosphere before it precipitates again. They found that evaporated water has the highest probability of staying in the atmosphere for around 5 days, but that there is a long tail of the probability distribution as many water particles still reside in the atmosphere more

than 25 days after evaporation, suggesting the mean to be much higher than the median.

In a near-global study van der Ent et al. (2014) extended the Eulerian moisture tracking model WAM-2layers to calculate both the amount and age of tracked water. Forcing their model with precipitation, wind and humidity from ERA-Interim (ERA-I) (Dee et al., 2011), and evaporation from the global hydrological model STEAM (Wang-Erlandsson et al., 2014) they computed atmospheric residence times over land. As such, they found that the residence time of land precipitation is

9.7 days and of precipitation recycled over land is 6.4 days. For evaporation they found a residence time of 8.7 days, with fast evaporation processes (interception, soil and open water evaporation) having a residence time of 8.1 days versus a residence time of 9.1 days for transpiration. In addition to these averages, they also showed spatial maps of residence times, but these refer to the fraction that recycles only, which is relatively fast compared to the average residence times. South America popped op as the continent with the lowest atmospheric residence times, which were shown to be around 4 days for moisture recycling

over land. As explained above, residence times of atmospheric moisture recycled above land are substantially lower than the

mean residence times of total moisture. The method of age tracers was previously also used by Numaguti (1999) who found residence times of multiple months. However, Numaguti (1999) started counting from sea evaporation in his experiments and continued the aging after water precipitated on land, infiltrated into the soil and re-evaporated. Therefore, his results cannot be interpreted as atmospheric residence times directly.

The established knowledge of the residence time of water in the atmosphere being 8–10 days was recently challenged by Läderach and Sodemann (2016). They used backward trajectories computed by the Lagrangian particle tracking method FLEXPART (forced with ERA-I) and concluded that the global mean residence time of water in the atmosphere is 4–5 days, i.e., half of the traditional estimates. More precisely, they calculated the residence time to be 3.9±0.8 days (spatial variability indicated by one standard deviation) for 15-day backward trajectories, 4.4 days for 20-day backward trajectories, and showed spatial figures of the first estimate. An obvious candidate for the discrepancy with the prevailing knowledge is the length of their trajectories, however, they stated that almost 100 % of the original moisture can be attributed to evaporation after 20 days, and that further backtracking is unphysical and can never come close to the 8–10 days estimate. Here, we would like to point out another important assumption, which was not addressed by Läderach and Sodemann (2016), namely that their methodology can accurately estimate evaporation. It should be known that FLEXPART is generally only used to look at specific humidity changes (evaporation – precipitation, or $E - P$). Attribution of evaporation is difficult as Sodemann et al. (2008) note that a number of moisture transport processes is neglected, which are moisture changes due to convection, turbulence, numerical diffusion, and rainwater evaporation. Stohl and Seibert (1998) even note that specific humidity fluctuations along a trajectory may be entirely unphysical. Stohl and James (2004), who evaluated the FLEXPART methodology, found that when FLEXPART is used to evaluate $E$ and $P$ separately, evaporation is highly overestimated. Specifically, they obtained a globally average evaporation of: $E = 1380 \, \mathrm{mm \, yr^{-1}}$, which corresponds to $E = 704 \cdot 10^3 \, \mathrm{km^3 \, yr^{-1}}$. Using the numbers obtained by Stohl and James (2004) and ERA-Interim atmospheric storage, a global average residence time of $12.4 \cdot 10^3 \, \mathrm{km^3}/704 \cdot 10^3 \, \mathrm{km^3 \, yr^{-1}} = 0.017 \, \mathrm{years} = 6.4 \, \mathrm{days}$ is obtained. Overestimation of evaporation will thus bias the estimates of residence times downward. How this assumption influences the results by LS16 was not evaluated in their paper. Whatever the methodological reason for the low global mean residence time estimates by Läderach and Sodemann (2016), we will argue in this paper that these numbers are physically impossible.

The objective of this paper is to revisit the current knowledge and provide a state-of-the-art view in time and space of the residence time of water in the atmosphere using several different approaches. Table 1 provides a non-exhaustive overview of the global average residence time of water in the atmosphere found by the studies mentioned in this introduction. Section 2 explains the methods used in this study. In Section 3 we explain why the global average residence time estimates based on quantifications of the global hydrological cycle are valid and the counterarguments provided by Läderach and Sodemann (2016) are not. In Section 4 we provide near-global spatial pictures of atmospheric residence time above land and sea. Section 5 analyzes the probability density function of residence time of atmospheric water particles, and finally, in Section 6 we state the conclusions of this paper.

**Table 1.** Non-exhaustive overview of (near-)global residence time estimates for water in the atmosphere in previous studies. Note that the estimates from this study are shown in Fig. 1.

| Study | Physical quantity estimated | Value (days) | Method |
| --- | --- | --- | --- |
| Chow et al. (1988) | Residence time (global average) | 8.2 | Global water balance |
| Jones (1997) | Residence time (global average) | 9.9 | Global water balance |
| Trenberth (1998) | Residence time (global average) | 9.1 | Global water balance ($P$-based) |
| Trenberth (1998) | Residence time (global average) | 8.9 | Global water balance ($E$-based) |
| Trenberth (1998) | Depletion time (spatial global average) | 8.1 | Local water balance ($P$-based) |
| Trenberth (1998) | Restoration time (spatial global average) | 8.5 | Local water balance ($E$-based) |
| Savenije (2000) | Residence time (global average) | 8.6 | Global water balance |
| Ward and Robinson (2000) | Residence time (global average) | 9.5 | Global water balance |
| Bosilovich and Schubert (2002) | Residence time (global average) | 8.5 | Online tracking method: tagged water depletion experiment (during May) |
| Bosilovich et al. (2002) | Residence time (global average) | 9.2 | Online tracking method: tagged water depletion experiment (period not specified) |
| Yoshimura et al. (2004) | Residence time (global average) | 7.3 | Offline tracking method: tagged water depletion experiment (April) |
| Yoshimura et al. (2004) | Residence time (global average) | 9.2 | Offline tracking method: tagged water depletion experiment (August) |
| Hendriks (2010) | Residence time (global average) | 10 | Global water balance |
| UCAR (2011) | Residence time (global average) | 9 | Global water balance |
| Bodnar et al. (2013) | Residence time (global average) | 9.5 | Global water balance |
| van der Ent et al. (2014) | Residence time (land $E$ only) | 8.7 | Offline tracking method: Eulerian age accounting |
| van der Ent et al. (2014) | Residence time (land $P$ only) | 9.7 | Offline tracking method: Eulerian age accounting |
| van der Ent et al. (2014) | Residence time ($P$ of land origin only) | 6.4 | Offline tracking method: Eulerian age accounting |
| Läderach and Sodemann (2016) | Residence time (spatial global average) | 8.3 | Local Eulerian method with transport |
| Läderach and Sodemann (2016) | Residence time (global average) | 3.9 | Offline tracking method: Lagrangian trajectories (15 days, for which all figures are presented) |
| Läderach and Sodemann (2016) | Residence time (global average) | 4.4 | Offline tracking method: trajectories (20 days) |
| Läderach and Sodemann (2016) | Residence time (global average) | 4–5 | Expert judgment based on tracking results and assumptions |

## 2 Methods

### 2.1 Global hydrological data

In this paper we use flux estimates of the global hydrological cycle as provided by Rodell et al. (2015). More specifically, we use the estimates of Rodell et al. (2015) that were optimized by forcing water and energy budget closure, taking into account uncertainty in the original estimates. For the estimation of precipitable water in the atmosphere we use the average of 8 reanalysis datasets provided by Trenberth et al. (2011) and we estimate uncertainty therein by calculating the variance of the 8 datasets assuming equal probability. The data describe the period of 2002–2008 (Trenberth et al., 2011) and the period of approximately the first decade of the 21th century (Rodell et al., 2015) respectively, but in the latter case it depends on the underlying input data. Moreover, we use global ERA-I precipitation and evaporation for the period 2002–2008 to compute the results based on ERA-I only.

### 2.2 Moisture tracking models

We use two different moisture tracking models, namely WAM-2layers (Water Accounting Model – 2 layers) (van der Ent, 2014, 2016) and 3D-T (3-Dimensional – Trajectories) (Tuinenburg, 2013) based on Dirmeyer and Brubaker (1999). These models are Eulerian and Lagrangian offline moisture tracking models respectively and we refer to their respective references for a detailed description. Both models track tagged water trough the atmosphere from its source (evaporation) to its sink (precipitation), or reversed when tracking precipitation back in time to the place where it evaporated. The most important assumption in both models is that precipitation stems from the entire atmospheric column (humidity weighted). Both models are improved from earlier versions and were validated against a detailed online moisture tracking method within a regional climate model (Knoche and Kunstmann, 2013; van der Ent et al., 2013). We force the models with global ERA-I data from which we use 2D fields of 3-hourly precipitation and evaporation, 6-hourly surface pressure, specific humidity and zonal and meridional wind speed on model levels covering the entire atmosphere from zero to surface pressure. Additionally, for 3D-T, we use 6-hourly vertical wind speeds on model levels. We use and present the data for the period 2002–2008, exactly corresponding to the period studied by Trenberth et al. (2011) and approximately the period studied by Rodell et al. (2015). For both models we use computational time steps of $0.25\,\mathrm{hour}$. For WAM-2layers we use 1 additional year at the beginning or end of this period for spin-up and a grid size of $1.5°$ latitude $\times$ $1.5°$ longitude. Vertical exchange between the two layers is parametrized and tagged water is not allowed to exceed the total water in the column. In the Lagrangian moisture tracking model we use a finer resolution, $0.25° \times 0.25°$, and we release one parcel every hour for each grid cell at a random horizontal location. The initial vertical location of a released parcel is 50 hPa above the land surface in the forward tracking scheme and humidity-weighted random in the backward tracking scheme. Some sensitivity tests on the number of parcels released showed that the release of one parcel per hour yielded stable results in terms of determining the residence time. Note that the validity of spatial and temporal variability in the results (Sections 4 and 5) depends strongly on how well ERA-I is able to describe the hydrological cycle.

### 2.2.1 Experiments with WAM-2layers

The model calculates the age $N_g$ of the tagged moisture present in a grid cell layer according to the following formula:

$$N_g^t = \frac{\left( \begin{array}{l} W_g^{t-1}\left(N_g^{t-1} + \Delta t\right) + \sum F_{g,\text{in}}\Delta t \left(N_{g,\text{in}}^{t-1} + \Delta t\right) \\ - \sum F_{g,\text{out}}\Delta t \left(N_g^{t-1} + \Delta t\right) - P_g \Delta t \left(N_g^{t-1} + \Delta t\right) + E_g \Delta t \frac{\Delta t}{2} \end{array} \right)}{W_g^t}, \tag{1}$$

where, the subscript $_g$ stands for tagged water. The superscripts $^t$ and $^{t-1}$ are the current and previous time step respectively. $\Delta t$ is the length of the time step. $N_{g,\text{in}}$ stands for the age of the tagged water coming into the grid cell layer. $F_{g,\text{in}}$ and $F_{g,\text{out}}$ are the incoming and outgoing fluxes over the (vertical and horizontal) boundaries of a grid cell layer.

With WAM-2layers we perform 4 experiments. In the first experiment we track continentally evaporated water over the globe until it precipitates and explicitly calculate the average age in each model layer of each grid cell at every time step. In the second experiment we do exactly the same, but now we backtrack continental precipitation to its origin. The third and fourth experiment are equal to the first and second, but now for tracking of oceanic evaporation and precipitation respectively.

The combined continental and oceanic tracking results are used to obtain global estimates (see Fig. 1) as well as spatial pictures of residence time and age of water in the atmosphere (see Figs. 2, 3, and S1). WAM-2layers does not work well with very small grid cells near the poles (time steps need to be very small to ensure stability, due to the explicit scheme used), thus we use data only between 80° S and 80° N (missing about 1 % of the Earth surface), and present our results between 75° S and 75° N (missing less than 3 % of the Earth surface). Tagged water crossing these boundaries is considered lost. For the global water balance calculations we do include data from the entire Earth, but evaporation and precipitation are very low at these high latitudes (about 0.2 % of the global hydrological cycle) and, thus, do not affect the results very much.

### 2.2.2 Experiments with 3D-Trajectories

With the Lagrangian moisture-tracking model, we perform a forward and a backward experiment. In the forward experiment, evaporation parcels from the Earth's surface are tracked through the atmosphere forward in time to their next precipitation location. In the backward experiment, precipitation parcels are tracked backward in time to their previous evaporation location.

Every time a moisture parcel (either evaporation or precipitation) is tracked forward or backward, its moisture balance is made during every time step. As a result of this moisture balance, we allocate a fraction of the original moisture to leave the atmospheric water cycle at that location. For example, in the forward experiment, we allocate a part of the evaporated water to each precipitation event located along the moisture trajectory. This procedure provides a probability density of the atmospheric residence time of the evaporation that is tracked forward or the precipitation that is tracked backward. The moisture is followed through the atmosphere during a period of 30 days, and, the residence time is accounted for the volume of moisture that leaves the atmosphere. After 30 days of tracking, if there is moisture that is unaccounted for, it is assumed to have a residence time of 30 days.

For each 0.25° grid cell (see Figs. 2, 3, and S1), these probability densities (see Fig. 4) are summed for all time steps during 2002–2008, weighted by the amount of evaporation or precipitation during the time of release. To acquire a global mean residence time, the global mean of local mean residence times are determined by weighting by total evaporation or precipitation volume (see Fig. 1).

## 3   Why the global average residence time of water in the atmosphere is 8–10 days

If one would like to know the average residence time $\tau$ in any reservoir one simply divides its average mass $\overline{M}$ (or volume when assuming constant density) by its average outgoing mass flux $\overline{F}$ (which equals the ingoing flux when there is no change of mass):

$$\tau = \frac{\overline{M}}{\overline{F}}. \tag{2}$$

Whereas this is a simple formula, computation of reliable residence times in, for example, a surface water lake may be difficult due to many uncertainties in a lake's volume, hydraulic flow, precipitation, evaporation and seepage (e.g., Monsen et al., 2002). Moreover, a lake may be permanently stratified (i.e. there is permanent dead storage) and one could argue that the actual volume participating in the water cycle of the lake does not equal the lake's total volume, meaning that the actual average residence time becomes lower. If one can, however, reliably estimate a lake's volume and in- or outflow, it is not necessary for a lake to be well-mixed for Eq. (2) to hold, nor is it necessary for $F$ to be constant. The mere necessity is that the entire volume participates in the water cycle. Of course, one could still have significant local differences, but the average can reliably be calculated by Eq. (2).

When the Earth's entire atmosphere is considered to be the reservoir of study, its residence time can actually be calculated much easier than that of a lake. In the global case the only inflow is evaporation and the only outflow is precipitation. Moreover, due to the turbulent nature of the atmosphere all water that resides in the atmosphere also participates in the atmospheric water cycle, i.e., there is no such thing as permanent dead water storage (e.g., Jacobson, 1999). This is furthermore supported by the online passive moisture tracking experiments of Bosilovich and Schubert (2002). Note that this online tracking method does not suffer from the well-mixed assumption for precipitation. Even so, they found that by tagging all moisture in the atmosphere, after 30 days, there was only 3 % of the tagged passive moisture left, thus at least 97 % of the total moisture in the atmosphere must have been participating in the hydrological cycle within 30 days.

The use of Eq. (2) to calculate the global mean residence time of atmospheric water has been recently criticized by Läderach and Sodemann (2016). They argued that Eq. (2) is a) not a reliable estimator for local residence times as it does not involve horizontal moisture transport, b) should be corrected for the surface are of the Earth where most precipitation is observed, and c) that the temporal characteristics of global precipitation cannot measured by depletion time constants. However, we disagree with these arguments as a*) horizontal moisture transport is irrelevant for the global average value, b*) the surface area of the Earth is irrelevant as it is not in Eq. (2), but nonetheless all areas participate in the hydrological cycle as there is also transport even over the Sahara (e.g.,  Goessling and Reick, 2013b; Schicker et al., 2010), and c*) the values in Eq. (2) correspond to

the elemental physical concept that an average residence time can be calculated from dividing a stock by its average influx or outflux under the assumption that there is negligible net stock change over a longer period; whether these average fluxes are constant or not is irrelevant, and the temporal characteristics of precipitation and evaporation can only affect the probability density functions of the residence time (Section 5), but not the average. In the Supplement we dispute the counterexamples objecting to Eq. (2) by Läderach and Sodemann (2016, Supplement Section 4) in more detail. Moreover, we argued above that the entire atmospheric volume participates in the hydrological cycle. Thus, Eq. (2) can, in our opinion, safely be used to calculate the global average residence time of atmospheric water.

Applying Eq. (2) on estimates of the global hydrological cycle (Fig. 1) yields a global mean residence time of atmospheric water of $8.9 \pm 0.4\,\mathrm{days}$ (uncertainty indicated by one standard deviation). The calculation of the mean is as follows:

$$\tau = \frac{12.6 \cdot 10^3}{(403.5 + 116.5) \cdot 10^3} = 0.024 \text{ years} = 8.9 \text{ days}, \tag{3}$$

and the standard deviation was calculated by general uncertainty propagation theory. The $1^{\text{th}}$ and $99^{\text{th}}$ percentile of this estimate are 7.9 and $9.8\,\mathrm{days}$ respectively. All previous global average estimates referred to in this paper roughly fall within this uncertainty range (Table 1), except for the estimate provided by Läderach and Sodemann (2016), which is less than half. Based on the arguments provided in this section we believe that the latter estimate is incorrect (the probability of an atmospheric residence time lower than $3.9\,\mathrm{days}$ equals $1 \cdot 10^{-30}$) . Even if we make the assumption that water outside the troposphere, thus in the stratosphere or mesosphere (where aerosol lifetimes in the order of $1\,\mathrm{year}$ have been found (Kristiansen et al., 2016)), does not participate in the hydrological cycle, the atmospheric storage is reduced by $\sim 1\,\%$ only. The corresponding global average residence time of water in the troposphere then is $8.8 \pm 0.4\,\mathrm{days}$. Thus, nowhere near the estimates by Läderach and Sodemann (2016). In the following example we show that their findings violate global mass balance: let us start from the fact that the average atmospheric water storage in ERA-I is $12.4 \cdot 10^3\,\mathrm{km^3}$ and average precipitation rate in ERA-I is $531 \cdot 10^3\,\mathrm{km^3\,yr^{-1}} = 1.45 \cdot 10^3\,\mathrm{km^3\,day^{-1}} = 2.85\,\mathrm{mm\,day^{-1}}$ (Fig. 1). For the sake of the example, let us neglect the $1\,\%$ moisture outside the troposphere. Then, the resulting active atmospheric storage is $12.3 \cdot 10^3\,\mathrm{km^3}$. For that atmospheric water to have a residence time of $3.9\,\mathrm{days}$, there must have been an evaporation rate of $12.3 \cdot 10^3 / 3.9 = 3.15 \cdot 10^3\,\mathrm{km^3\,day^{-1}} = 6.03\,\mathrm{mm\,day^{-1}}$, which is physically impossible as that means an enormous imbalance in $P$ and $E$. Alternatively, it would require an enormous part of the atmospheric water to be dead storage (i.e., never participate in the hydrological cycle), namely $12.4 \cdot 10^3 - 1.45 \cdot 10^3 \times 3.9 = 6.7 \cdot 10^3\,\mathrm{km^3}$ or $54\,\%$ of all water in the atmosphere.

Figure 1 shows several estimates of global residence times. ERA-I fluxes fall well within the uncertainty ranges provided by Rodell et al. (2015), but they are generally on the high side. Thus, the global mean residence time estimate of $8.4$–$8.6\,\mathrm{days}$, based on global ERA-I data, is slightly on the low side of the uncertainty spectrum. The estimates from our moisture tracking methods (WAM-2layers and 3D-T), which use ERA-I, match quite well for global mean atmospheric residence time. The moisture tracking estimates split out for land and ocean are, therefore, slightly lower than the most likely value of $8.9\,\mathrm{days}$. However, it is clear that the turnover of water in the atmosphere is faster over the ocean than over land, having a difference of about $2\,\mathrm{days}$. The logical explanation here is that the hydrological cycle over the ocean is not limited by dry conditions on land, and, as a consequence, is more intense.

## 4 A spatial view on the residence time of atmospheric moisture

Figure 2 provides a near-global spatial view of the annual average hydrological cycle, atmospheric residence times and age.
Globally averaged, precipitation residence time, evaporation residence time and age of atmospheric water should be the same, but may differ due to imbalances in the data of the atmospheric hydrological cycle (see Fig. 1). More importantly, these three metrics have a different physical meaning, and thus, a different spatial pattern (Figs. 2c–e). Let us consider a particular location in the world, in this case Portugal, as an example. As can be seen from Fig. 2d, moisture which evaporates from Portugal stays in the atmosphere on average about 14–15 days before it rains out again. In other words: the atmospheric residence time of evaporation is 14–15 days. The local recycling of atmospheric water is only a few percent (e.g., Dirmeyer et al., 2009; van der Ent and Savenije, 2011), and much of the evaporated atmospheric moisture is, in fact, transported towards relatively dry regions in the Mediterranean and Africa (e.g., Schicker et al., 2010; van der Ent et al., 2010), hence the relatively long atmospheric residence time of evaporation. On the other hand, the precipitation in Portugal comes for a large part from oceanic sources relatively nearby (e.g., Dirmeyer et al., 2009; Gimeno et al., 2012; van der Ent and Savenije, 2013), and we estimate that is has resided in the atmosphere for about 7–8 days (Fig. 2c) before it fell as precipitation in Portugal. In other words: the atmospheric residence time of precipitation is 7–8 days. The spatial image of the age of atmospheric water (Fig. 2e) is very similar to the precipitation residence time (Fig. 2c). For our Portugal example, the average age of atmospheric water is about about 8–10 days. Precipitation draws its water from the atmospheric reservoir with a certain age, but apparently, the atmospheric moisture in the drier months has a higher age. Hence, for Portugal, the time-averaged age of atmospheric moisture can be somewhat higher than the precipitation-weighted atmospheric residence time of precipitation.

With Fig. 2 we would like to stress that there are multiple ways of looking at the residence of atmospheric moisture. Whether you look at residence time from a precipitation perspective (Fig. 2c) or an evaporation perspective (Fig. 2d) gives an entirely different picture. In the precipitation perspective, the time from the previous evaporation is stressed, while in the evaporation perspective, the time to the next precipitation event is stressed. The definition of an atmospheric residence time – for both precipitation and evaporation – is analogous to the definition of other metrics for the atmospheric branch of the hydrological cycle, which are also defined for both the precipitation and evaporation perspectives (e.g., Trenberth, 1998, 1999; van der Ent et al., 2010; van der Ent and Savenije, 2011). Moreover, you can also look at the actual age of atmospheric water as it resides in the atmosphere (Fig. 2e). Figure 2f provides the latitudinal averages of Figs. 2c–e, as well as of the estimates from WAM-2layers and 3D-T separately (Fig S1), for which a discussion is attached in the Supplement.

Figure 2c is directly comparable to a recent estimate (Läderach and Sodemann, 2016, Fig. 2a). Their spatial patterns are very similar, however, we observe that they underestimate the residence time everywhere with a factor 2–3 compared to our results. As pointed out in the introduction, the method of Läderach and Sodemann (2016) relies on the unevaluated assumption that they can accurately attribute evaporation, and on a rather short length of their trajectories (15 or 20 days). In contrast, our methods use longer trajectories (WAM-2layers: continuous; 3D-T: 30 days), and use the fields of ERA-I evaporation directly. Moreover, our global average values fit with the atmospheric water balance (Eq. (2) and Fig. 1). For an extensive discussion of

the counterarguments of Läderach and Sodemann (2016) against the use of Eq. (2), and our rebuttal, we would like to refer the reader to the Supplement attached to this paper.

We make the following observations based on Fig. 2:

- The places of low precipitation residence times (Fig. 2c) coincide mostly with areas of low precipitation (Fig. 2a). This indicates that if there is precipitation its water content has recently evaporated and is most likely of local origin. Note that the reverse statement: low precipitation (Fig. 2a) coinciding with low precipitation residence times (Fig. 2c) is not necessarily true;

- The intertropical convergence zone (ITCZ) has increasing precipitation residence times (Fig. 2c) and decreasing evaporation residence times (Fig. 2d) towards its center. This holds over the ocean as well as over the northern Amazon and Indonesia. The atmospheric residence time of evaporation (Fig. 2d) can often be seen as an indication of the moisture travel distance towards an area of high precipitation such as the ITCZ;

- The southern hemisphere ocean (roughly between $45° \text{S}$–$75° \text{S}$) has evaporation residence times (Fig. 2d) of less than $5 \, \text{days}$, while absolute evaporation is low (Fig. 2b). This can be explained by relatively high precipitation rates (Fig. 2a) compared to the amount of atmospheric water vapor that is in the air (e.g., NASA Earth Observatory, 2016), so evaporation has a quick turnover.

- Over the Sahara, age of atmospheric water (Fig. 2e) as well as residence times (Fig. 2c,d) are more than $15 \, \text{days}$, indicating that moisture comes from remote sources. Over the Tibetan Plateau there is a reversed situation where the age of atmospheric water and residence times are low relative to surrounding values.

- When following the atmospheric moisture flow inland from the coast, the age of atmospheric water increases (Fig. 2e), as does the residence time of precipitation. This feature is very clear over Eurasia, but can be observed on other continents as well.

- Latitudinally averaged (Fig. 2f), the precipitation residence time peaks towards the poles and the equator. This is in anticorrelation with the residence time of evaporation. This corresponds with the Hadley and Ferrel cells, transporting evaporated atmospheric moisture from the high-pressure zones with the prevailing trade winds and westerlies towards areas of high precipitation.

Figure 3 shows the age and residence times of atmospheric water in January and July. In January (Fig. 3a) the age of atmospheric water is relatively low in the Northern Hemisphere, and relatively high in the Southern Hemisphere. In July (Fig. 3c) the pattern is reversed. In much of the Southern Hemisphere the atmospheric moisture storage (see Supplement Fig. S3) is lower in July (Fig. S3b) compared to January (Fig. S3a), precipitates rates are higher in July (Fig. S3d) compared to January (Fig. S3c), and evaporation rates or also higher in July (Fig. S3f) compared to January (Fig. S3e). The higher evaporation rates in July in the ERA-I data may seem counterintuitive, but correspond to previous studies (e.g., Yu, 2007). It is, therefore, quite logical that lower storage and higher fluxes lead to lower moisture ages in the Southern Hemisphere in July

(Fig. 3c) compared to January (Fig. 3a). Note that Animation 1 in the Supplement provides a view of moisture age throughout the year. The seasonal patterns we observe for precipitation residence times (Fig. S2a+c) are quite similar to Läderach and Sodemann (2016, Supplement Fig. S3), albeit that their figures are for DJF and JJA while ours are for January and July, and, as for the yearly average figures, their absolute values are much lower.

Over the continents, especially Eurasia, the pattern of atmospheric water age is complex and we observe an interesting ocean-to-land contrast (Fig. 3). In January, looking at the Northern Hemisphere, the ocean is relatively warm compared to the land and the age of atmospheric water increases going inland, as there is little replenishment from land evaporation. In July the opposite situation occurs: the ocean is relatively cool compared to the land and the age of atmospheric water decreases going inland. This corresponds to high evaporation rates and corresponding high continental moisture recycling ratios (e.g., Dirmeyer et al., 2014; van der Ent, 2014). As a result, the atmospheric water age over a large part of Asia is actually lower in July compared to January. Over the Southern Hemisphere we see the same ocean-to-land contrast, but it is less pronounced as there is relatively little land present. The latitudinally averaged residence times of precipitation and evaporation (Fig. 3b,d) also show the summer vs. winter reversal.

## 5   The probability density function of the residence time of atmospheric moisture

Figure 4 shows probability density functions (PDFs) for residence times. Theoretically, the global PDFs of evaporation and precipitation residence time should be identical. However, they slightly differ, which can be attributed to inconsistencies in the ERA-I forcing data as well as the assumptions made in the tracking model. Regarding the forcing data, it seems from Fig. 1 that the lifetime of atmospheric moisture is slightly too short, a common issue in all models (Trenberth et al., 2003), indicating that our results may be slightly skewed towards lower residence times. According to Trenberth et al. (2003), precipitation also falls too early in the day in all models, thus the amplitude of residence times over land could also be affected, but it is unclear to what extent. Regarding the modeling assumptions, in 3D-T we assume a humidity-weighted well-mixed atmosphere during precipitation and the starting location of the trajectories is randomized over the grid cell. These assumption may lead to an underestimation of the number of water particles that undergo a very fast cycle, and may, thus, slightly skew our results towards higher residence times. By definition of mass balance, however, the actual mean of the distribution should not change. When more water particles undergo a faster(slower) cycle, as a logical consequence, also more water particles undergo a slower(faster) cycle. Adding age tracers to online tracking methods (Wei et al., 2016), but then applied to new global methods (e.g., Singh et al., 2016), would allow to check the validity and consequences of these assumptions in more detail, however, would still depend on the model world.

Looking at Fig. 4 we see that short residence times have the highest probability, but there is a long tail with low probabilities and high residence times. We find the median residence time of precipitation and evaporation to be 5.7 and 4.6 days respectively. Thus, the median is about 3 to 4 days less than the mean (see Fig. 1), indicating that the long tails skew the mean significantly. About 5 % of the moisture has residence times of more than 30 days., which we assumed to have a residence time of 30 days, when we calculated the mean (Fig. 1). As a consequence, the estimates for the mean from 3D-T may be slightly

lower than the 'true' values. It is unclear how Läderach and Sodemann (2016) have dealt with the unattributed moisture after the end of their trajectories, but they already attributed 97 % of the initial precipitation to evaporation after just 15 days in their Lagrangian model. In Section 3, however, we argued that this is physically impossible from a global mass balance perspective.

We furthermore observe an interesting daily cycle in the residence time PDFs over land (Fig. 4), while the general shape of ocean and land PDFs is not very different when looking at timescales of multiple days. We suggest that the daily cooling and warming, resulting in a daily cycle of land evaporation, and a higher likelihood of precipitation to occur towards the end of the day, causes the daily cycle in the residence time PDFs. This phenomenon is only visible over land, as here surface cooling and warming occurs with much greater amplitude than over the ocean. These cycles are still visible after multiple days (Fig. 4).

## 6 Conclusions

In this paper we studied the residence time of water in the atmosphere. We revisited the state-of-the-art knowledge and studied its properties in time and space on a global scale. As discussed in previous sections, our results are naturally limited by the validity of the input data and the assumptions in our tracking models. However, we trust our results enough to draw the following main conclusions:

- Given the state-of-the-art estimates of the hydrological cycle, the global mean residence time of atmospheric water is $8.9 \pm 0.4$ days (uncertainty indicated by one standard deviation). The $1^{\text{th}}$ and $99^{\text{th}}$ percentile of this estimate are 7.9 and 9.8 days respectively;

- The average atmospheric residence time over the ocean is about 2 days lower than over land;

- Locally, there are different perspectives of looking at residence time. Atmospheric residence time of evaporation can often be seen as an indication of the moisture travel distance towards an area of high precipitation such as the ITCZ, while atmospheric residence time of precipitation is often more complex;

- Latitudinally averaged, residence time of precipitation peaks towards the poles and the equator, which is in anticorrelation with the residence time of evaporation;

- In winter, the age of atmospheric water is generally several days lower than in summer;

- In the Northern Hemisphere, following atmospheric moisture inland, the age of atmospheric water increases when the sea is relatively warm compared to the land and decreases when the sea is relatively cold. This cannot clearly be observed in the Southern Hemisphere, where less continental mass is present;

- Probability density functions of atmospheric residence time have long tails, with a global median of around 5 days.

*Author contributions.* R.J.v.d.E. and O.A.T. designed the study, performed the analysis and wrote the paper.

*Acknowledgements.* We would like to acknowledge ECMWF for supplying ERA-Interim data through their server at www.ecmwf.int. We thank Niek van de Koppel and Tolga Cömert from Delft University of Technology for the translation of the code of WAM-2layers from Matlab to Python, which has been used for this study. Furthermore, we thank Patrick Keys of Stockholm University for comments on the HESSD version of the paper. Moreover, we would like to thank the editor and everyone that has participated in the lively interactive discussion. R.J.v.d.E. received funding from the European Union Seventh Framework Programme (FP7/2007–2013) under grant agreement no. 603608, Global Earth Observation for integrated water resource assessment: eartH2Observe. The views expressed herein are those of the authors and do not necessarily reflect those of the European Commission.

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

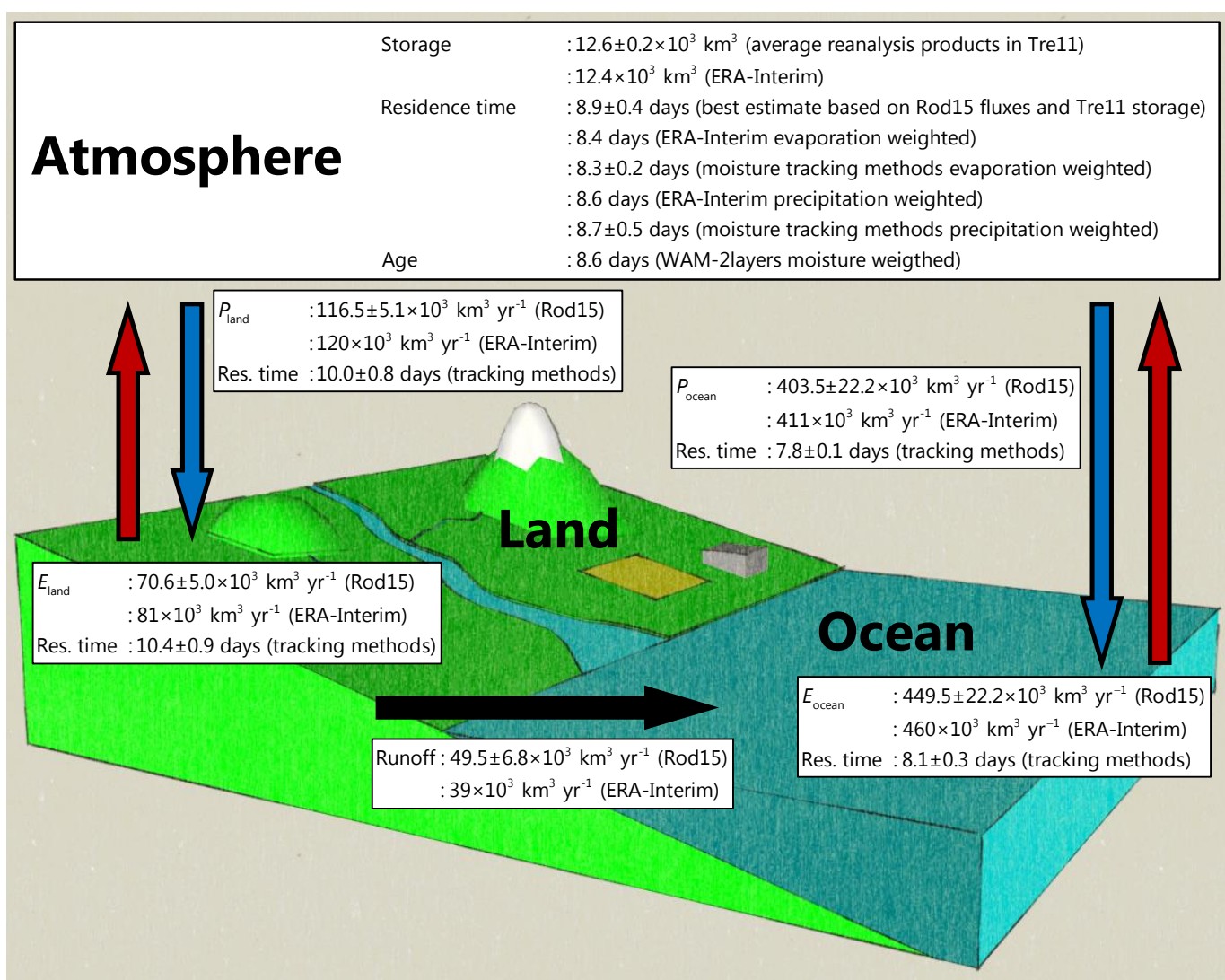

**Figure 1.** Earth's hydrological cycle with residence times (2002–2008). All residence times shown are weighted averages. The uncertainty ranges indicated for the residence times from the moisture tracking methods refer to the uncertainty associated with model choice (WAM-2layers or 3D-Trajectories). Tre11 stands for Trenberth et al. (2011) and Rod15 stands for Rodell et al. (2015). The land area is $147 \cdot 10^6\ \mathrm{km}^2$ and the ocean area is $363 \cdot 10^6\ \mathrm{km}^2$.

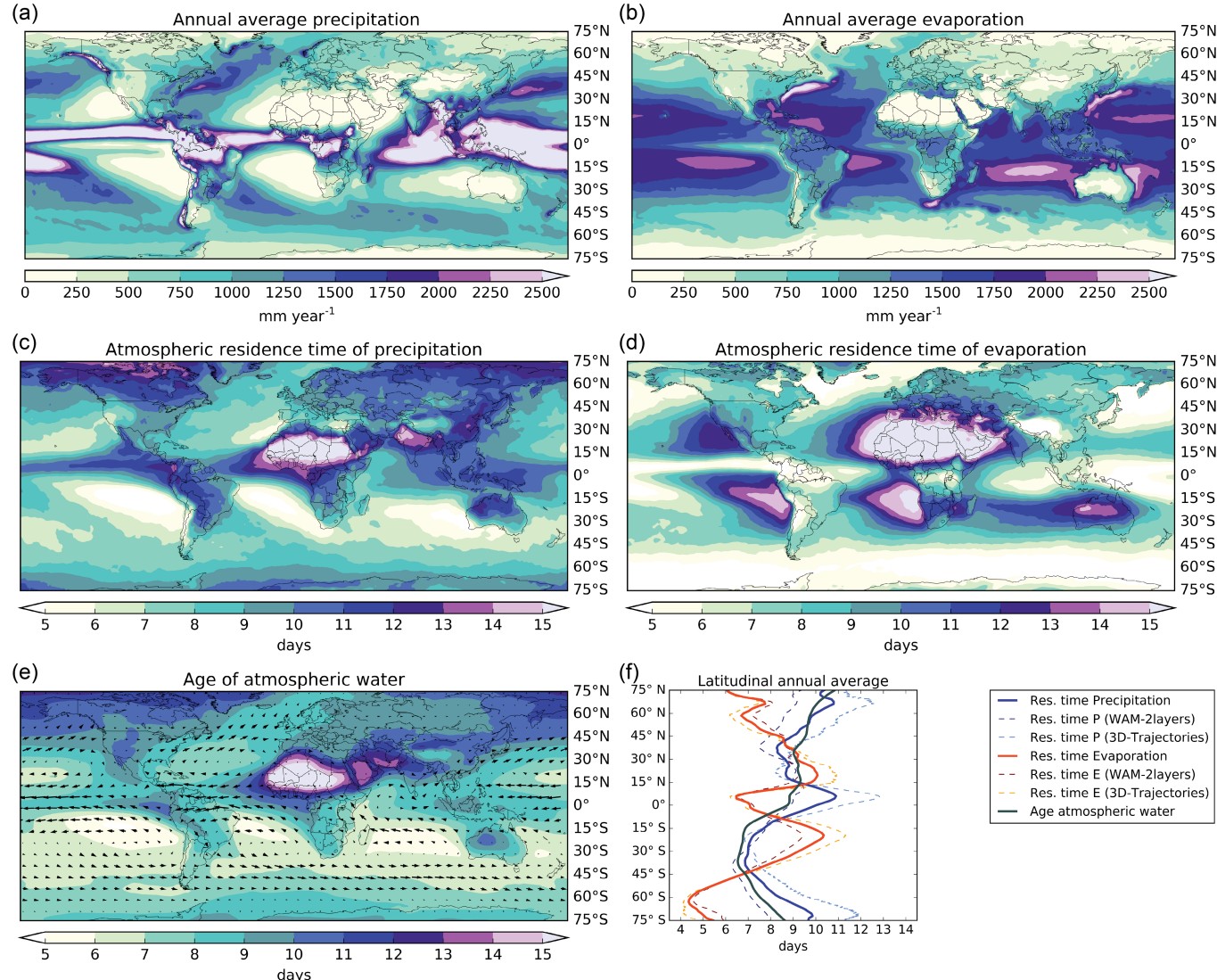

**Figure 2.** Annual average hydrological cycle, atmospheric residence times and age for 2002–2008, based on ERA-Interim data. **(a)** Precipitation, **(b)** evaporation, **(c)** weighted average atmospheric residence time of precipitation (average of WAM-2layers and 3D-T), **(d)** weighted average atmospheric residence time of precipitation (average of WAM-2layers and 3D-T), and **(e)** time averaged age of atmospheric water as it is the atmospheric column (WAM-2layers). The arrows indicate the vertically integrated moisture fluxes. **(f)** Latitudinal averages. The individual estimates from WAM-2layers and 3D-T for panels (c) and (d) can be found in the Supplement (Fig. S1). The age of atmospheric water (e) for all days individually can also be found in the Supplement (Animation 1).

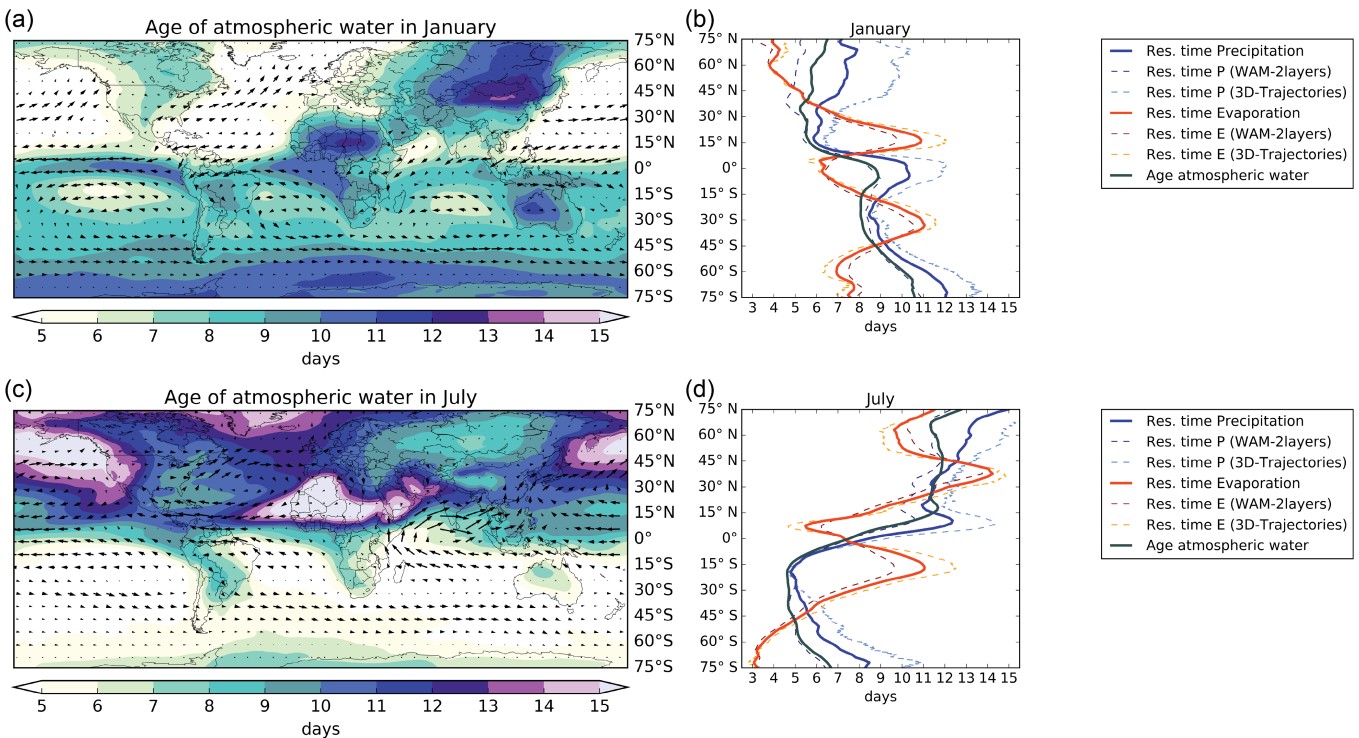

**Figure 3.** Time metrics in January and July. **(a)** Time-averaged age of atmospheric water in January (2002–2008) as computed by WAM-2layers based on ERA-Interim data. The arrows indicate the vertically integrated moisture fluxes. **(b)** Latitudinal averages of residence times and water age in January. **(c)** As (a) for July. **(d)** As (b) for July. The spatial residence time figures for January and July can be found in the Supplement (Fig. S2).

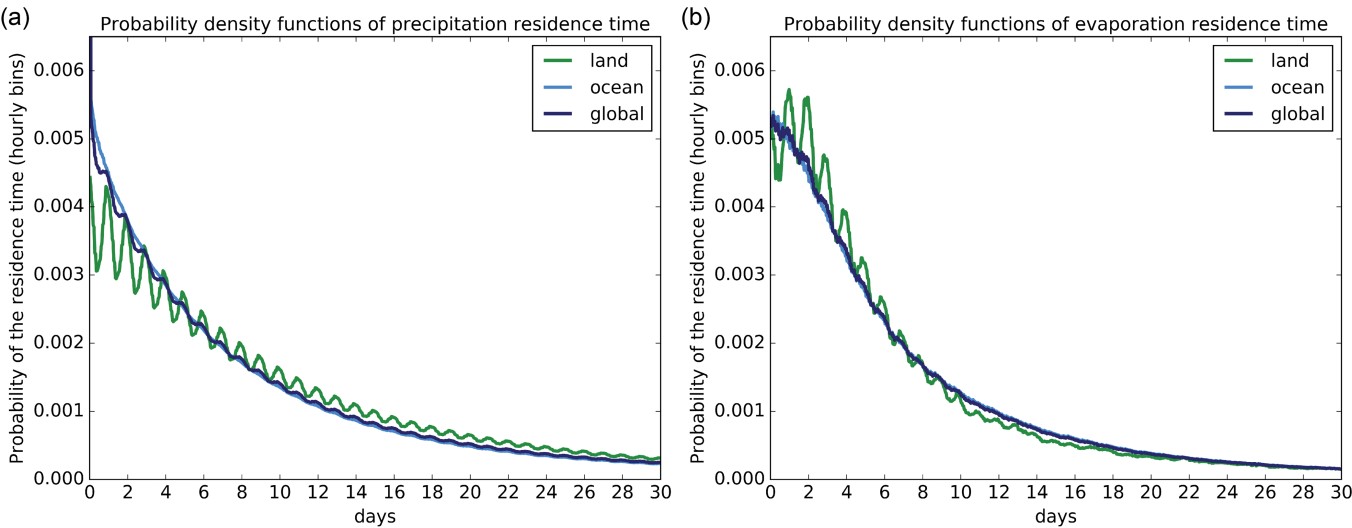

**Figure 4.** Probability density functions (PDFs) of atmospheric residence time as computed by 3D-Trajectories based on ERA-Interim data (2002–2008). **(a)** PDFs of precipitation residence time, and **(b)** PDFs of evaporation residence time. About 5 % of the moisture has residence times of more than 30 days.