# Peer review of "The residence time of water in the atmosphere revisited"

_Hydrology and Earth System Sciences, 2016_

## Referee Comment (RC1) · K. Trenberth (Referee) · 6 Sep 2016

Review comments on: HESS Title: The residence time of water in the atmosphere revisited Author(s): R. J. van der Ent and O. A. Tuinenburg MS No.: hess-2016-431 MS Type: Research article

Comments by Kevin E. Trenberth, NCAR 4 Sept 2016

This paper addresses the issue of residence time of atmospheric moisture and concludes that the value originally proposed by Trenberth (1998) of 8.9 days still applies. It provides a partial commentary on an earlier paper by Läderach and Sodemann (2016) which suggested that the residence time was less than half, namely 3.9 days. Two methods are used to assess the lifetime and age of moisture in the atmosphere and the basic fields used come from ERA-Interim reanalyses.

[Figure]

Although some issues in addressing these scientific questions are discussed, many outstanding issues and reasons for different results are not. The tracking models used in this paper deal with particles and not finite volumes; and hence they do not appear to deal with the water budgets and precipitation processes or the storms and how they reach out to gather in moisture. Tracking a parcel is not the same as tracking the overall moisture flow from source to sink. Mixing and convection do not appear to be dealt with and "precipitation events" are not defined. These processes are not reversible (one can not go backwards, but in this paper they do). It is not that the exercise in this paper is without merit, but rather that it involves huge unstated assumptions and many questions are left outstanding.

Evaporation, as the source of moisture, is continuous and rates are modest. In contrast, precipitation is inherently intermittent; it typically precipitates only about 7 to 10% of the time (depends on threshold), and the precipitation processes vary enormously. Most precipitation occurs in the Tropics and is convective in nature, and this is generally true in summer over continents as well. Weather systems are typically much smaller in scale in summer over land than in winter where large extratropical baroclinic storms provide the main storms. None of these aspects are addressed in this paper. Atmospheric and climate models, including high-resolution numerical weather prediction models, have grid scales of tens to hundreds of km, and convection is parameterized. It has been shown in many studies that precipitation in models occurs too frequently and with insufficient intensity owing to the convective parameterizations, so that the lifetime of moisture in the atmosphere is much too short in all models. The easiest way to show this is via the strong summer diurnal cycle in precipitation and it's timing, which is too early in the day in all models (see Trenberth et al. 2003 for a discussion of all these points.).

Because precipitation rates (when raining) average 10 to 25 times evaporation rates (see Trenberth et al. 2003) (owing to the fact that most of the time it does not rain), any moderate or intense precipitation comes from advection and convergence of water

vapor, not local evaporation. Monsoons are an example where moisture is transported great distances in reality to supply the monsoon rains, and a chronic error in most models is that the precipitation is deficient in monsoon areas (see Christensen et al. 2013), because the moisture falls out prematurely.

The difficulty of dealing with precipitation processes realistically, and especially convection, is a major outstanding issue in all studies that address the lifetime of moisture in the atmosphere. No doubt the problems in the Läderach and Sodemann (2016) paper stem from these issues. The methods applied in this paper do not appear to suffer from the premature onset of convection because they do not deal with realistic precipitation processes at all!

Results should be reconciled with estimates of "recycling" of moisture, which refers to the amount of moisture over a particular area that is precipitated from evaporation within that area (see Trenberth 1999). That paper also discusses and presents estimates of the older concepts of "intensity of the hydrological cycle", "precipitation efficiency" and "moistening efficiency" which have unfortunately been lost in this paper.

There remain major issues also in the datasets used in all such studies. Here, the evaporation and precipitation are from ERA-interim, which is a model-based assimilated set of values. Over land, evaluation of precipitation using Global Precipitation Climatology Centre (GPCC) high resolution data (Becker et al. 2013) shows considerable shortcomings in the reanalysis values (Schneider et al., 2013); also Trenberth et al. (2011). Globally, the Global Precipitation Climatology Project (GPCP) analyses are most widely accepted as having best values, although these are monthly means. Evaporation analyses suffer from shortcomings associated with bulk flux estimates, and are only useful in the context of a complete water cycle (as in Rodell et al. 2015).

Some further questions that arose for me are as follows. It makes no sense to me to separately compute a "precipitation residence time" and "evaporation residence time". Perhaps they should be called something else (e.g. see Trenberth 1999)? It states

"The places of low precipitation residence times (Fig. 2c) coincide mostly with areas of low precipitation (Fig. 2a)." Yet if it does not rain, then perhaps moisture hangs around for a long time? It seems count-intuitive? Or is it because in subtropical high pressure systems perhaps the moisture is transported away? "The intertropical convergence zone (ITCZ) has increasing precipitation residence times" yet it is pouring with rain? Is this because the moisture has been transported from afar? Isn't the age of atmospheric moisture dependent on the precipitation processes? Several of the results here related to regional residence times also do not appear to make sense from a standpoint of the physical process associated with precipitation and the water cycle. The seasonal differences over the southern hemisphere in Figure 3 are surprising to say the least (I am from New Zealand), and seem very suspicious elsewhere too (such as over the northern ocean storm tracks). Extratropical storms are every bit as active in summer in the southern hemisphere as they are in winter, just for a narrower latitudinal band (Trenberth 1991). The results cry out for explanations.

There appear to be problems in Eq (1) since it deals with t, t-1, and $\Delta$t. The units are inconsistent because "1" has no units.

Added references

Becker, A., P. Finger, A. Meyer‐Christoffer, B. Rudolf, K. Schamm, U. Schneider and M. Ziese, 2013. A description of the global land‐surface precipitation data products of the Global Precipitation Climatology Centre with sample applications including centennial (trend) analysis from 1901–present. Earth Syst. Sci. Data, 5, 71‐99, doi: 10.5194/essd‐5‐71‐2013 Christensen, J.H., K. Krishna Kumar, E. Aldrian, S.-I. An, I.F.A. Cavalcanti, M. de Castro, W. Dong, P. Goswami, A. Hall, J.K. Kanyanga, A. Kitoh, J. Kossin, N.-C. Lau, J. Renwick, D.B. Stephenson, S.-P. Xie and T. Zhou, 2013: Climate Phenomena and their Relevance for Future Regional Climate Change. In: Climate Change 2013: The Physical Sci ňence Basis. Contribution of Working Group I to the Fifth Assessment Report of the Intergovernmental Panel on Climate Change [Stocker, T.F., et al. (eds.)]. Cambridge University Press, Cambridge, U. K.

and New York, NY, USA. Schneider, U., A. Becker, P. finger, A. Meyer-Christoffer, M. Ziese, and B. Rudolf, 2014: GPCC's new land surface precipitation climatology based on quality-controlled in situ data and its role in quantifying the global water cycle. Theor. Appl. Climatol. 115, 15-40. Trenberth, K. E., 1991: Storm tracks in the southern hemisphere. J. Atmos. Sci., 48, 2159–2178. Trenberth, K. E., 1999: Atmospheric moisture recycling: Role of advection and local evaporation. J. Climate, 12, 1368-1381. Trenberth, K. E., A. Dai, R. M. Rasmussen and D. B. Parsons, 2003: The changing character of precipitation. Bull. Amer. Meteor. So
* * *

---

## Author Comment (AC1) · 14 Sep 2016

**The residence time of water in the atmosphere revisited**

Ruud J. van der Ent[1] and Obbe A. Tuinenburg[2]

[1]Department of Physical Geography, Faculty of Geosciences, Utrecht University, Utrecht, the Netherlands
[2]Department of Environmental Sciences, Copernicus Institute for Sustainable development, Utrecht University, Utrecht, the Netherlands

*Correspondence to:* Ruud J. van der Ent (r.j.vanderent@uu.nl)

**Response to the review of Kevin Trenberth**

We thank referee Kevin Trenberth for the prompt review of our manuscript. We agree with several of the comments, however, we argue here that some comments regarding our data, methods and metrics are not very constructive, not true or not relevant considering the scope of our paper. Nonetheless, it is our job to clearly outline the objectives of this paper, justify the methods

5 used and explain our results. Therefore, we will adjust the revised version of the manuscript wherever appropriate, but first we provide a detailed response here below. Comments by the referee are in italic and replies are in normal text. The detailed adjustments to the revised manuscript will follow after the public discussion period.

**General comments**

10 *This paper addresses the issue of residence time of atmospheric moisture and concludes that the value originally proposed by Trenberth (1998) of 8.9 days still applies. It provides a partial commentary on an earlier paper by Läderach and Sodemann (2016) which suggested that the residence time was less than half, namely 3.9 days. Two methods are used to assess the lifetime and age of moisture in the atmosphere and the basic fields used come from ERA-Interim reanalyses.*

Indeed, the referee is correct that we conclude that the traditional estimate for global average residence time of water in

15 the atmosphere is 8.9±0.4 days. However, we would like to point out that it is hard to speak of a single original estimate as Trenberth (1998) actually provided two estimates, namely 8.9 days and 9.1 days, depending on whether global average evaporation or precipitation is considered. Moreover, the earliest reference in our manuscript is in fact to Chow et al. (1988), which suggested 8.2 days, but in Läderach and Sodemann (2016) one can find references to even earlier estimates.

20 *Although some issues in addressing these scientific questions are discussed, many outstanding issues and reasons for different results are not.*

It is not entirely clear to us what the referee means with this general and non-constructive comment. Our paper comprehensively discusses many aspects related to the residence time of water in the atmosphere, which was the main objective of the paper.

25

*The tracking models used in this paper deal with particles and not finite volumes; and hence they do not appear to deal with the water budgets and precipitation processes or the storms and how they reach out to gather in moisture. Tracking a parcel is*

*not the same as tracking the overall moisture flow from source to sink. Mixing and convection do not appear to be dealt with and "precipitation events" are not defined. These processes are not reversible (one cannot go backwards, but in this paper they do). It is not that the exercise in this paper is without merit, but rather that it involves huge unstated assumptions and many questions are left outstanding.*

5    There appear to be a few big misunderstandings here. The first model (Water Accounting Model – 2 layers) actually keeps account of the tagged and total water volumes in two layers of each grid cell globally and thus explicitly deals with water budgets. The second model (3-Dimensional – Trajectories) uses many water parcels, which in our definition are infinitesimal water volumes, to represent the total moisture budget of the atmospheric column. Both models are forced with climate data (ERA-Interim in this case), which does contain the dynamics of precipitation events and storms, as that will be reflected in

10   the precipitation, humidity and wind data. Vertical mixing, however, is indeed a difficult issue to tackle for all atmospheric moisture tracking models and is likely to be a major cause for differences between WAM-2layers and 3D-T. The assumptions involved and effect on the outcome are comprehensively discussed elsewhere (van der Ent, 2014; van der Ent et al., 2013; Tuinenburg, 2013). In contrast to some other models, for example FLEXPART, which tracks (E-P) parcels (Stohl et al., 2005), the two models we used in this research do track atmospheric moisture from source to sink, which is clearly explained in their

15   respective references (van der Ent, 2014; Tuinenburg, 2013). Backward tracking of atmospheric moisture surely comes with assumptions, but it is widely applied in so-called offline moisture tracking models, see e.g., Gimeno et al. (2012) for a non-exhaustive overview, and additionally Keys et al. (2012). Unless the referee makes the "many outstanding questions"' explicit we cannot react to this comment. We propose the following changes to our manuscript:

  – In Section 2.2, add a sentence where we explicitly state that both models track moisture from source to sink;

20   – Add a paragraph to the Supplement that discusses the differences between WAM-2layers and 3D-T (Fig. S1) in terms of their underlying assumptions.

*Evaporation, as the source of moisture, is continuous and rates are modest. In contrast, precipitation is inherently intermittent; it typically precipitates only about 7 to 10 % of the time (depends on threshold), and the precipitation processes vary enormously. Most precipitation occurs in the Tropics and is convective in nature, and this is generally true in summer over*

25   *continents as well. Weather systems are typically much smaller in scale in summer over land than in winter where large extratropical baroclinic storms provide the main storms. None of these aspects are addressed in this paper.*

We totally agree with the observations made by the referee regarding evaporation and precipitation. However, for as far as the differences between evaporation and precipitation are relevant for atmospheric residence time we feel that they are already discussed in Section 4.

30

*Atmospheric and climate models, including high-resolution numerical weather prediction models, have grid scales of tens to hundreds of km, and convection is parameterized. It has been shown in many studies that precipitation in models occurs too frequently and with insufficient intensity owing to the convective parameterizations, so that the lifetime of moisture in the*

*atmosphere is much too short in all models. The easiest way to show this is via the strong summer diurnal cycle in precipitation and it's timing, which is too early in the day in all models (see Trenberth et al. 2003 for a discussion of all these points.).*

Again, we totally agree with the observations made by the referee. Most likely the ERA-Interim data is also affected by having too frequent and not intense enough rainfall, but a detailed investigation of this is beyond the scope of this paper. We believe that the largest effect on our findings would be on the probability density functions of residence time (Fig. 4). Namely, we think that the amplitude of the diurnal cycle that we observed could be slightly higher in this case. We propose the following change to our manuscript:

- In Section 5, add a sentence about a likely bias in precipitation frequency and intensity in our data, with reference to Trenberth et al. (2003), and how that could potentially (slightly) increase the amplitude of the diurnal cycle observed in Fig. 4.

*Because precipitation rates (when raining) average 10 to 25 times evaporation rates (see Trenberth et al. 2003) (owing to the fact that most of the time it does not rain), any moderate or intense precipitation comes from advection and convergence of water vapor, not local evaporation. Monsoons are an example where moisture is transported great distances in reality to supply the monsoon rains, and a chronic error in most models is that the precipitation is deficient in monsoon areas (see Christensen et al. 2013), because the moisture falls out prematurely.*

We again absolutely agree. Unfortunately, the referee does not make clear how he thinks these comments are related to our paper. If it is to say that our forcing data is uncertain then he is of course right, but this is an issue of any paper that uses data. We also address uncertainty in this paper, but we do not see the need to repeat over and over again that our results depend on uncertain forcing. We think that most, if not all, readers of HESS are very much aware of all uncertainties associated with climate and weather data, or, in fact, any type of data.

*The difficulty of dealing with precipitation processes realistically, and especially convection, is a major outstanding issue in all studies that address the lifetime of moisture in the atmosphere. No doubt the problems in the Läderach and Sodemann (2016) paper stem from these issues. The methods applied in this paper do not appear to suffer from the premature onset of convection because they do not deal with realistic precipitation processes at all!*

We use ERA-Interim forcing data just as Läderach and Sodemann (2016) do. Obviously, the tracking methods are different as we come up with a very different estimate. Yet, the spatial patterns are quite similar as discussed on P6:L28-30 of our manuscript. In contrast, however, to the estimates of Läderach and Sodemann (2016), our estimates actually makes sense from a global average water budget calculation (see Section 3). In our opinion, the comment that we do not deal with realistic precipitation processes at all is unfounded and inappropriate.

*Results should be reconciled with estimates of "recycling" of moisture, which refers to the amount of moisture over a particular area that is precipitated from evaporation within that area (see Trenberth 1999). That paper also discusses and presents*

*estimates of the older concepts of "intensity of the hydrological cycle", "precipitation efficiency" and "moistening efficiency" which have unfortunately been lost in this paper.*

We agree that recycling of moisture is indeed an important aspect when studying the time component of atmospheric water. However, we would like to point out that recycling of moisture has much broader definitions than the one proposed by Trenberth (1999). See, for example, earlier work on more regional scales (Brubaker et al., 1993; Eltahir and Bras, 1994; Savenije, 1995; Schär et al., 1999), but also later work (e.g., Bisselink and Dolman, 2008; Dominguez et al., 2006; Tuinenburg et al., 2012) , which place recycling in a local, regional and continental context. On a global scale, recycling has also been discussed extensively in previous papers from us as well as from others using the same, or similar methods, as the ones used here (e.g., Dirmeyer et al., 2009, 2014; van der Ent and Savenije, 2011, 2013; van der Ent et al., 2010, 2014) and some of these papers added the perspective of evaporation recycling. The concepts of precipitation efficiency and moistening efficiency, as globally calculated by Trenberth (1999), are surely useful as well for some purposes, but they remain local metrics, which are not helpful for our research objectives as stated on P3:L15-16: "The objective of this paper is to revisit the current knowledge and provide a state-of-the-art view in time and space of the residence time of water in the atmosphere". Läderach and Sodemann (2016) explain quite well why local metrics are not good estimates of residence time.

*There remain major issues also in the datasets used in all such studies. Here, the evaporation and precipitation are from ERA-interim, which is a model-based assimilated set of values. Over land, evaluation of precipitation using Global Precipitation Climatology Centre (GPCC) high resolution data (Becker et al. 2013) shows considerable shortcomings in the reanalysis values (Schneider et al., 2013); also Trenberth et al. (2011). Globally, the Global Precipitation Climatology Project (GPCP) analyses are most widely accepted as having best values, although these are monthly means. Evaporation analyses suffer from shortcomings associated with bulk flux estimates, and are only useful in the context of a complete water cycle (as in Rodell et al. 2015).*

Surely the people of GPCC and GPCP are doing a tremendous job in providing as good as possible precipitation estimates. Whether those estimates are the best is quite arbitrary, although we agree that they are probably better than ERA-Interim. However, ERA-I precipitation is given in a 3-hourly resolution and is most consistent with the other data from ERA-I. As can be seen from Fig. 1, ERA-I falls within the uncertainty ranges estimated by Rodell et al. (2015) on a global scale, as is discussed on P6:L16-20. As mentioned before, we think that most, if not all, readers of HESS are very much aware of all uncertainties associated with data of the climate, or, in fact, any type of data, without the necessity of repeating this over and over again.

**Specific comments**

Some further questions that arose for me are as follows. It makes no sense to me to separately compute a "precipitation residence time" and "evaporation residence time". Perhaps they should be called something else (e.g. see Trenberth 1999)? The fact that Fig. 2c is so different from Fig. 2d illustrates that it makes a lot of sense to define an atmospheric residence time for both precipitation and evaporation. This is actually analogous to defining both depletion and restoration times (Trenberth, 1998), precipitation and moistening efficiency (1999), or precipitation and evaporation recycling metrics (van der Ent and

Savenije, 2011; van der Ent et al., 2010). However, atmospheric residence times are clearly different from all of these metrics, thus it is necessary to define them. We propose the following change to our manuscript:

- In Section 4, related to Figs. 2c,d, we will mention that the definition of an atmospheric residence time – for both precipitation and evaporation – is analogous to the definition of other metrics for the atmospheric branch of the hydrological cycle, which are also defined for both the precipitation and evaporation perspectives (e.g., Trenberth, 1998, 1999; van der Ent and Savenije, 2011; van der Ent et al., 2010).

*It states "The places of low precipitation residence times (Fig. 2c) coincide mostly with areas of low precipitation (Fig. 2a)." Yet if it does not rain, then perhaps moisture hangs around for a long time? It seems count-intuitive? Or is it because in subtropical high pressure systems perhaps the moisture is transported away?*

Our observation is exactly as it reads here, and not the other way around, i.e., areas of low precipitation do not always coincide with areas of low precipitation residence time. In the Sahara, for example, atmospheric moisture indeed hangs around for a long time without being replenished with evaporation. We propose the following change to our manuscript:

- Add a remark that the reverse statement (i.e., low precipitation coinciding with low precipitation residence time) is not necessarily true.

*"The intertropical convergence zone (ITCZ) has increasing precipitation residence times" yet it is pouring with rain? Is this because the moisture has been transported from afar? Isn't the age of atmospheric moisture dependent on the precipitation processes?*

Indeed the moisture has been transported from afar, this is somewhat more clear in Fig. 2d than in Fig. 2c, and, therefore, we wrote on P7:L5-6 "The atmospheric residence time of evaporation (Fig. 2d) can often be seen as an indication of the moisture travel distance towards an area of high precipitation such as the ITCZ". The age of atmospheric moisture is not dependent on the precipitation processes. When you take away water from any reservoir with a certain age, the age itself is not directly influenced, but only when it is again mixed with new water.

*Several of the results here related to regional residence times also do not appear to make sense from a standpoint of the physical process associated with precipitation and the water cycle. The seasonal differences over the southern hemisphere in Figure 3 are surprising to say the least (I am from New Zealand), and seem very suspicious elsewhere too (such as over the northern ocean storm tracks). Extratropical storms are every bit as active in summer in the southern hemisphere as they are in winter, just for a narrower latitudinal band (Trenberth 1991). The results cry out for explanations.*

Unfortunately, the referee is not very specific about what he thinks is suspicious in our results. It is not clear to us whether he would expect lower or higher atmospheric moisture ages in the Southern Hemisphere during January or July and in which latitudinal band specifically. Below we added Figure S3, which may be able to provide some clarification. In much of the Southern Hemisphere the atmospheric moisture storage (i.e., precipitable water) is lower in July (Fig. S3b) compared to January (Fig. S3a), precipitates rates are higher in July (Fig. S3d) compared to January (Fig. S3c), and evaporation rates or also higher

in July (Fig. S3f) compared to January (Fig. S3e). The higher evaporation rates in July in the ERA-Interim data may seem counterintuitive, but correspond to previous studies (e.g., Yu, 2007). Is it quite logical that lower storage and higher fluxes lead to lower moisture ages in the Southern Hemisphere in July (Fig. 3c) compared to January (Fig. 3a). Note that Animation 1 in the Supplement provides a nice view of moisture age throughout the year. Moreover, the seasonal patterns we observe for precipitation residence times (Fig. S2a+c in the Supplement) are quite similar to Läderach and Sodemann (2016, Fig S3 in the Supplement), albeit that their figures are for DJF and JJA while ours are for January and July, and, as for the yearly average figures, their absolute values are much lower. We propose the following change to our manuscript:

- Add Figure S3 here below to the Supplement and provide more explanation similar to the explanation above in Section 4 (around the discussion of Fig. 3).

**Technical corrections**

*There appear to be problems in Eq. (1) since it deals with t, t-1, and t. The units are inconsistent because "1" has no units.*

We thank the referee for noting this. We will rewrite the equation to avoid possible confusion as follows:

The model calculates the age $N_g$ of the tagged moisture present in a grid cell layer according to the following formula:

$$N_g^t = \frac{\left( \begin{array}{l} W_g^{t-1}\left(N_g^{t-1}+\Delta t\right) + \sum F_{g,\text{in}}\Delta t\left(N_{g,\text{in}}^{t-1}+\Delta t\right) \\ -\sum F_{g,\text{out}}\Delta t\left(N_g^{t-1}+\Delta t\right) - P_g\Delta t\left(N_g^{t-1}+\Delta t\right) + E_g\Delta t\dfrac{\Delta t}{2} \end{array} \right)}{W_g^t},$$
(1)

where, the subscript $g$ stands for tagged water. The superscripts $t$ and $t-1$ are the current and previous time step respectively. $\Delta t$ is the length of the time step. $N_{g,\text{in}}$ stands for the age of the tagged water coming into the grid cell layer. $F_{g,\text{in}}$ and $F_{g,\text{out}}$ are the incoming and outgoing fluxes over the (vertical and horizontal) boundaries of a grid cell layer.

**References**

Bisselink, B. and Dolman, A. J.: Precipitation recycling: Moisture sources over Europe using ERA-40 data, J. Hydrometeorol., 9(5), 1073–1083, 2008.

Brubaker, K. L., Entekhabi, D. and Eagleson, P. S.: Estimation of continental precipitation recycling, J. Clim., 6(6), 1077–1089, 1993.

Dirmeyer, P. A., Brubaker, K. L. and DelSole, T.: Import and export of atmospheric water vapor between nations, J. Hydrol., 365(1-2), 11–22, doi:10.1016/j.jhydrol.2008.11.016, 2009.

Dirmeyer, P. A., Wei, J., Bosilovich, M. G. and Mocko, D. M.: Comparing Evaporative Sources of Terrestrial Precipitation and Their Extremes in MERRA Using Relative Entropy, J. Hydrometeorol., 15(1), 102–106, doi:10.1175/JHM-D-13-053.1, 2014.

Dominguez, F., Kumar, P., Liang, X. Z. and Ting, M.: Impact of atmospheric moisture storage on precipitation recycling, J. Clim., 19(8), 1513–1530, 2006.

Eltahir, E. A. B. and Bras, R. L.: Precipitation recycling in the Amazon Basin, Q. J. R. Meteorol. Soc., 120(518), 861–880, 1994.

Gimeno, L., Stohl, A. and Trigo, R.: Oceanic and terrestrial sources of continental precipitation, Rev. Geophys., 50, RG4003, doi:10.1029/2012RG000389, 2012.

Keys, P. W., van der Ent, R. J., Gordon, L. J., Hoff, H., Nikoli, R. and Savenije, H. H. G.: Analyzing precipitationsheds to understand the vulnerability of rainfall dependent regions, Biogeosciences, 9(2), 733–746, doi:10.5194/bg-9-733-2012, 2012.

Läderach, A. and Sodemann, H.: A revised picture of the atmospheric moisture residence time, Geophys. Res. Lett., 43, 924–933, doi:10.1002/2015GL067449, 2016.

Rodell, M., Beaudoing, H. K., L'Ecuyer, T. S., Olson, W. S., Famiglietti, J. S., Houser, P. R., Adler, R., Bosilovich, M. G., Clayson, C. A., Chambers, D., Clark, E., Fetzer, E. J., Gao, X., Gu, G., Hilburn, K., Huffman, G. J., Lettenmaier, D. P., Liu, W. T., Robertson, F. R., Schlosser, C. A., Sheffield, J. and Wood, E. F.: The observed state of the water cycle in the early twenty-first century, J. Clim., 28(21), 8289–8318, doi:10.1175/JCLI-D-14-00555.1, 2015.

Savenije, H. H. G.: New definitions for moisture recycling and the relationship with land-use changes in the Sahel, J. Hydrol., 167(1-4), 57–78, 1995.

Schär, C., Lüthi, D., Beyerle, U. and Heise, E.: The soil-precipitation feedback: A process study with a regional climate model, J. Clim., 12(2-3), 722–741, 1999.

Stohl, A., Forster, C., Frank, A., Seibert, P. and Wotawa, G.: Technical note: The Lagrangian particle dispersion model FLEXPART version 6.2, Atmos. Chem. Phys., 5(9), 2461–2474, doi:10.5194/acp-5-2461-2005, 2005.

Trenberth, K. E.: Atmospheric moisture residence times and cycling: Implications for rainfall rates and climate change, Clim. Change, 39(4), 667–694, doi:10.1023/A:1005319109110, 1998.

Trenberth, K. E.: Atmospheric moisture recycling: Role of advection and local evaporation, J. Clim., 12(5 II), 1368-1381, 1999.

Trenberth, K. E., Dai, A., Rasmussen, R. M. and Parsons, D. B.: The changing character of precipitation, Bull. Am. Meteorol. Soc., 84(9), 1205-1217+1161, 2003.

Tuinenburg, O. A.: Atmospheric Effects of Irrigation in Monsoon Climate: The Indian Subcontinent, Wageningen University. [online] Available from: http://edepot.wur.nl/254036, 2013.

Tuinenburg, O. A., Hutjes, R. W. A. and Kabat, P.: The fate of evaporated water from the Ganges basin, J. Geophys. Res., 117(D1), D01107, doi:10.1029/2011jd016221, 2012. van der Ent, R. J.: A new view on the hydrological cycle over continents, Delft University of Technology., doi:10.4233/uuid:0ab824ee-6956-4cc3-b530-3245ab4f32be, 2014. van der Ent, R. J. and Savenije, H. H. G.: Length and time scales of atmospheric moisture recycling, Atmos. Chem. Phys., 11(5), 1853–1863, doi:10.5194/acp-11-1853-2011, 2011.

van der Ent, R. J. and Savenije, H. H. G.: Oceanic sources of continental precipitation and the correlation with sea surface temperature, Water Resour. Res., 49(7), 3993–4004, doi:10.1002/wrcr.20296, 2013.

van der Ent, R. J., Savenije, H. H. G., Schaefli, B. and Steele-Dunne, S. C.: Origin and fate of atmospheric moisture over continents, Water Resour. Res., 46(9), W09525, doi:10.1029/2010WR009127, 2010.

35     van der Ent, R. J., Tuinenburg, O. A., Knoche, H. R., Kunstmann, H. and Savenije, H. H. G.: Should we use a simple or complex model for moisture recycling and atmospheric moisture tracking?, Hydrol. Earth Syst. Sci., 17(12), 4869–4884, doi:10.5194/hess-17-4869-2013, 2013.

    van der Ent, R. J., Wang-Erlandsson, L., Keys, P. W. and Savenije, H. H. G.: Contrasting roles of interception and transpiration in the hydrological cycle - Part 2: Moisture recycling, Earth Syst. Dyn., 5(2), 471–489, doi:10.5194/esd-5-471-2014,
5   2014.

    Yu, L.: Global Variations in Oceanic Evaporation (1958-2005): The Role of the Changing Wind Speed, J. Clim., 20(21), 5376–5390, doi:doi:10.1175/2007JCLI1714.1, 2007.

[Figure]

**Figure S3.** Atmospheric moisture storage, precipitation and evaporation in January and July (2002—2008, ERA-Interim).

---

## Referee Comment (RC2) · K. Trenberth (Referee) · 16 Sep 2016

The fundamental issue is the whole methodology used by this and other studies, which the evidence suggests does not give the correct answers for reasons given. Namely the processes are not reversible, precipitation processes are not addressed (they are parameterized in models), and the lifetime of moisture in models is too short). Indeed precipitation processes are parameterized in the ECMWF model. The study does not deal with realistic precipitation processes. I am fine with the authors saying what their assumptions are and here are the results, but it is another matter to claim they represent the real world.

---

## Referee Comment (RC3) · J. Wei (Referee) · 19 Sep 2016

This paper revisited the issue of atmospheric moisture residence time, especially the estimation from an earlier study, by using two different models, one Eulerian and one Lagrangian. They argue that the estimation from the earlier study is not correct. The methods are sophisticated, but I feel that some issues are not clear to me. I hope the authors can clarify them and make the paper earlier to understand.

I do not understand why the residence time estimated from a precipitation perspective, an evaporation perspective, and the age of atmospheric water are different (Fig. 2c-2e). Do they indicate the same physical characteristic? Are the differences caused by the different methods and imbalance of the hydrological data?

In the top of page 6, you criticized Läderach and Sodemann (2016) by arguing that

horizontal moisture transport is irrelevant for the global average residence time. I think Läderach and Sodemann (2016) showed results of both with and without moisture transport, and both of them are about half of 8 days. So it is not clear to me what is main problem of the study of Läderach and Sodemann (2016) that leads to the estimated low residence time if your paper and their paper are talking about the same physical quantity (e.g., there are difference between the residence time and depletion time constants as shown in Läderach and Sodemann (2016)).

Other specific comments:

In the introduction, you reviewed many past studies on the residence time. It will be more clear and organized if you use a table to list all the residence time values.

Page 2, Line 3. "local moisture feedback" is not clear here and needs more explanation.

Page 2, Line 14-15. "No details were given whether these experiments were performed in summer or winter." This statement is hard to believe for published papers.

Page 4, Eq.(1). Why the last term "Eg dt dt/2" is different from other flux terms?

Page 6, Line 18, There should be a comma after "EAR-I data"

Page 6, Line 20. "the most likely value". What is it?

Page 6, line 26-27. "In the precipitation perspective, the time from the previous evaporation is stressed, while in the evaporation perspective, the time to the next precipitation event is stressed." Can you clearly explain what this means?

Page 8, Lines 4 and 17. Can you give some explanation why the residence time over the ocean is about 2 days lower than over land?

Page 7, Line 9. "amount of atmosphere"?

---

## Referee Comment (RC4) · J. Wei (Referee) · 28 Sep 2016

I thank the authors for carefully addressing my comments. There is one thing still not clear.

As to the dramatically different results on moisture residence time from your estimates and those from Läderach and Sodemann (2016), what do you think is the main reason? Do they have some problems/errors in science or technology? For example, is there any problem with the FLEXPART model or data sets they used? Such large difference in results should not come from some trivial differences in model and data.

---

## Author Comment (AC2) · 28 Sep 2016

This is the response to the response of the referee Kevin Trenberth. The comment by the referee is in italic and the reply in normal text.

*The fundamental issue is the whole methodology used by this and other studies, which the evidence suggests does not give the correct answers for reasons given. Namely the processes are not reversible, precipitation processes are not addressed (they are parameterized in models), and the lifetime of moisture in models is too short). Indeed precipitation processes are parameterized in the ECMWF model. The study does not deal with realistic precipitation processes. I am fine with the authors saying what their assumptions are and here are the results, but it is another matter to claim they represent the real world.*

We thank the referee for pointing out that precipitation processes are parameterized in the ECMWF model and thus the ERA-Interim data. However, any model is an attempt to represent the real world, and any model fails in doing that exactly, because it is naturally different from the real world. In our opinion, we have clearly stated our assumptions and presented our results under these assumptions. We be no means intended to overstate our results, but we have built and expanded on previous research, and, therefore, think we have made a relevant contribution to the scientific literature. We also nowhere in the paper use terms like "truth" or "reality", but our results are of course an attempt to say something meaningful about the real world, given the state-of-the-art data and models at hand.

---

## Author Comment (AC3) · 28 Sep 2016

**The residence time of water in the atmosphere revisited**

Ruud J. van der Ent[1] and Obbe A. Tuinenburg[2]

[1]Department of Physical Geography, Faculty of Geosciences, Utrecht University, Utrecht, the Netherlands
[2]Department of Environmental Sciences, Copernicus Institute for Sustainable development, Utrecht University, Utrecht, the Netherlands

*Correspondence to:* Ruud J. van der Ent (r.j.vanderent@uu.nl)

**Response to the review of Jiangfeng Wei**

We thank Jiangfeng Wei for the positive and constructive review. Comments by the referee are in italic and replies are in normal text. The detailed adjustments to the revised manuscript will follow after the public discussion period.

**General comments**

*This paper revisited the issue of atmospheric moisture residence time, especially the estimation from an earlier study, by using two different models, one Eulerian and one Lagrangian. They argue that the estimation from the earlier study is not correct. The methods are sophisticated, but I feel that some issues are not clear to me. I hope the authors can clarify them and make the paper easier to understand.*

We will clarify the issues, pointed out by the referee, in our responses below and in the revised version of the manuscript.

*I do not understand why the residence time estimated from a precipitation perspective, an evaporation perspective, and the age of atmospheric water are different (Fig. 2c- 2e). Do they indicate the same physical characteristic? Are the differences caused by the different methods and imbalance of the hydrological data?*

Globally averaged, precipitation residence time, evaporation residence time and age of atmospheric water should be the same. Indeed, estimates may differ due to imbalances in the data of the atmospheric hydrological cycle. However, these three metrics also have a different physical meaning, and thus, a different spatial pattern (Figs. 2c–e). Let us consider a particular location in the world, let say Portugal, as an example. As can be seen from Fig. 2d, moisture which evaporates from Portugal stays in the atmosphere on average about 14–15 days before it rains out again. In other words: the atmospheric residence time of evaporation is 14–15 days. The local recycling of atmospheric water is only a few percent (e.g., Dirmeyer et al. 2009; van der Ent and Savenije, 2011), and much of the evaporated atmospheric moisture is, in fact, transported towards relatively dry regions in the Mediterranean and Africa (e.g., Schicker et al., 2010; van der Ent et al., 2010), hence the relatively long atmospheric residence time of evaporation. On the other hand, the precipitation in Portugal comes for a large part from oceanic evaporative sources relatively nearby (e.g., Dirmeyer et al. 2009; Gimeno et al., 2012; van der Ent and Savenije, 2013), and we estimate that is has resided in the atmosphere for about 7–8 days (Fig. 2c) before it fell as precipitation in Portugal. In other words: the atmospheric residence time of precipitation is 7–8 days. The spatial image of the age of atmospheric water (Fig. 2e) is very similar to the precipitation residence time (Fig. 2c). For our Portugal example, the average age of atmospheric water

is about about 8–10 days. Precipitation draws its water from the atmospheric reservoir with a certain age, but apparently, the atmospheric moisture in the drier months has a higher age. Hence, for Portugal, the time averaged age of atmospheric moisture can be somewhat higher than the precipitation weighted atmospheric residence time of precipitation. We hope that this issue will be made clear by the following change to our manuscript:

5     – In Section 4, we will use a shortened version of the Portugal example above to clarify the differences between the three metrics displayed in Figs. 2c–e.

*In the top of page 6, you criticized Läderach and Sodemann (2016) by arguing that horizontal moisture transport is irrelevant for the global average residence time. I think Läderach and Sodemann (2016) showed results of both with and without moisture transport, and both of them are about half of 8 days. So it is not clear to me what is main problem of the study of Läderach*

10 *and Sodemann (2016) that leads to the estimated low residence time if your paper and their paper are talking about the same physical quantity (e.g., there are difference between the residence time and depletion time constants as shown in Läderach and Sodemann (2016)).*

We derive a global average residence time of atmospheric water of $8.9\pm0.4\,\mathrm{days}$ (uncertainty given as one standard deviation), whereas Läderach and Sodemann (2016) derive this to be $3.9\pm0.8\,\mathrm{days}$ (spatial variability indicated by one standard

15 deviation) for 15-day backward trajectories. These estimates are clearly different. The controversy is, in fact, not in the depletion time constants (with or without the transport approximation) as we agree with that Läderach and Sodemann (2016) that depletion time constants are different from actual residence times.

**Specific comments**

20 *In the introduction, you reviewed many past studies on the residence time. It will be more clear and organized if you use a table to list all the residence time values.*

We thank the referee for this suggestion. Initially, we were a bit hesitant to include such a table because a table could suggest completeness, whereas there are most likely more textbooks, general water papers and educational web pages that include an estimate of the global average residence time. Moreover, several other numbers consider quantities which are a bit different

25 as they are depletion times, or consider residence times only above land or only of recycled moisture. However, to increase readability we will follow the referee's suggestion to include a table in the manuscript with the following headers: Study – Physical quantity estimated – Value – Method. We will add a note that the table is non-exhaustive.

*Page 2, Line 3. "local moisture feedback" is not clear here and needs more explanation.*

30 We intend to change the wording here to: "However, it is safer to interpret them as local time scales of atmospheric moisture recycling (van der Ent and Savenije, 2011)".

*Page 2, Line 14-15. "No details were given whether these experiments were performed in summer or winter." This statement is hard to believe for published papers.*

We have checked the references again. In Bosilovich and Schubert (2002) it appears that the experiment in question was performed in May, but for Bosilovich et al. (2002) we could not find when this experiment was exactly performed. The latter reference is in fact a publication in GEWEX News and not a publication in a journal. We will update the revised manuscript accordingly.

*Page 4, Eq.(1). Why the last term "$E_g \Delta t \frac{\Delta t}{2}$" is different from other flux terms?*

Because fluxes are per unit of time these are all multiplied by the time step $\Delta t$. Next, they are multiplied by the age at timestep $t$. Thus:

Flux * time step * age.

For the outgoing fluxes the age at time step $t$ is given by $(N_g^{t-1} + \Delta t)$. At the time a water particle evaporates its age is actually 0, however, we assume evaporation uniformly distributed over $\Delta t$, thus the resulting age of evaporated water from time step $t-1$ to time step $t$ is $\frac{\Delta t}{2}$.

*Page 6, Line 20. "the most likely value". What is it?*

This concerns the value in Eq. (3) of $8.9 \, \text{days}$. We will specify this between brackets in the revised version.

*Page 6, line 26-27. "In the precipitation perspective, the time from the previous evaporation is stressed, while in the evaporation perspective, the time to the next precipitation event is stressed." Can you clearly explain what this means?*

See the response above concerning Figs. 2c–e.

*Page 8, Lines 4 and 17. Can you give some explanation why the residence time over the ocean is about 2 days lower than over land?*

The atmospheric hydrological cycle is apparently more intense over the ocean than over land as indicated also at the end of Section 3. We will add this in the revised version.

**Technical corrections**

*Page 7, Line 9. "amount of atmosphere"?*

This should have read "amount of atmospheric water"

*Page 6, Line 18, There should be a comma after "ERA-I data"*

OK

**References**

Bosilovich, M. G. and Schubert, S. D.: Water vapor tracers as diagnostics of the regional hydrologic cycle, J. Hydrometeorol., 3(2), 149–165, doi:10.1175/1525-7541(2002)003<0149:WVTADO>2.0.CO;2, 2002.

Bosilovich, M. G., Sud, Y., Schubert, S. D. and Walker, G. K.: GEWEX CSE sources of precipitation using GCM water vapor tracers, GEWEX News, 12(3), 1,6–7,12, 2002.

Dirmeyer, P. A., Brubaker, K. L. and DelSole, T.: Import and export of atmospheric water vapor between nations, J. Hydrol., 365(1-2), 11–22, doi:10.1016/j.jhydrol.2008.11.016, 2009.

Gimeno, L., Stohl, A. and Trigo, R.: Oceanic and terrestrial sources of continental precipitation, Rev. Geophys., 50, RG4003, doi:10.1029/2012RG000389, 2012.

Läderach, A. and Sodemann, H.: A revised picture of the atmospheric moisture residence time, Geophys. Res. Lett., 43, 924–933, doi:10.1002/2015GL067449, 2016.

Schicker, I., Radanovics, S. and Seibert, P.: Origin and transport of Mediterranean moisture and air, Atmos. Chem. Phys., 10(11), 5089–5105, doi:10.5194/acp-10-5089-2010, 2010.

van der Ent, R. J. and Savenije, H. H. G.: Length and time scales of atmospheric moisture recycling, Atmos. Chem. Phys., 11(5), 1853–1863, doi:10.5194/acp-11-1853-2011, 2011.

van der Ent, R. J. and Savenije, H. H. G.: Oceanic sources of continental precipitation and the correlation with sea surface temperature, Water Resour. Res., 49(7), 3993–4004, doi:10.1002/wrcr.20296, 2013.

van der Ent, R. J., Savenije, H. H. G., Schaefli, B. and Steele-Dunne, S. C.: Origin and fate of atmospheric moisture over continents, Water Resour. Res., 46(9), W09525, doi:10.1029/2010WR009127, 2010.

---

## Author Comment (AC4) · 4 Oct 2016

The comment from the referee is in italic and our response is in normal text

*I thank the authors for carefully addressing my comments. There is one thing still not clear. As to the dramatically different results on moisture residence time from your estimates and those from Läderach and Sodemann (2016), what do you think is the main reason? Do they have some problems/errors in science or technology? For example, is there any problem with the FLEXPART model or data sets they used? Such large difference in results should not come from some trivial differences in model and data.*

The referee raises an important issue here. The large differences between the

global average number – for atmospheric residence time – obtained with FLEXPART (Läderach and Sodemann, 2016) compared to our two tracking models (WAM-2layers and 3D-Trajectories) should not come from trivial differences between the models. We showed in our supplementary material that WAM-2layers and 3D-Trajectories also have small differences, but nowhere near the differences both models have with FLEXPART (in absolute terms, spatial patterns actually quite similar). Most importantly, the global average residence time computed with our models is close to the numbers computed from a simple global water balance, namely 8–10 days. We think that this is an indication of some sort of big error in the analysis with FLEXPART (Läderach and Sodemann, 2016). However, having not worked with the FLEXPART model and its input data ourselves it is difficult to pinpoint this error. We prefer to refrain from speculation at this point, but we have invited the Alexander Läderach and Harald Sodemann the join the discussion here and hopefully that will shed some light on the issue.

**References**

Läderach, A. and Sodemann, H.: A revised picture of the atmospheric moisture residence time, Geophys. Res. Lett., 43, 924–933, doi:10.1002/2015GL067449, 2016.

---

## Referee Comment (RC5) · Anonymous Referee #3 · 13 Oct 2016

This paper looks to address the question regarding the residence time of water in the atmosphere recently raised by Laderach and Sodemann (2016). I feel this study does not provide enough evidence to significantly contribute to the discussion.

1) There are large limitations to using tracking models to track atmospheric moisture. The authors need to go into a lot more detail as to how the models were applied to this dataset. Specifically, how the water in the model is tagged, how this tagged water relates to the evaporation and precipitation in the subsequent time step, whether mixing of this water is taken into account. Without going into more detail as to how the tracking models specifically deal with water it is not possible to address question of the residence time of the water.

2) The authors highlight the difference in the assumptions made by Laderach and Sodermann (2016), but more discussion as to why the authors disagree with these differences is needed. No evidence is provided to support the authors' choice in assumptions over the choice by Laderach and Sodermann.

3) The analysis of the results conclude that ERA-Interim shows close agreement to the previous studies of residence time, however does not provide a response to Laderach and Sodermann.

The paper needs to go into more detail of the tracking methods, a more detailed analysis of the results and a greater discussion regarding the assumptions made by Laderach and Sodermann, and to provide evidence as to why the authors disagree with these assumptions.

---

## Short Comment (SC1) · 19 Oct 2016

**Comment on van der Ent and Tuinenburg, "The residence time of water in the atmosphere revisited"**
*Harald Sodemann, 20.10.2016*

Until recently, it was commonly accepted knowledge that the residence time of water vapour should be about 8-10 days on a global average. In our recent study published in Geophysical Research Letters (Läderach and Sodemann, 2016, henceforth LS16) we challenged this viewpoint with regard to different aspects. First, we pointed out deficiencies in the use of depletion time constants if used as a local measure of the time water vapour resides in the atmosphere between evaporation and precipitation. From a modified Eulerian estimate taking into account horizontal moisture flux between grid cells, we derive more plausible patterns that are in correspondence with prevailing weather systems. Using a Lagrangian method that identifies the sources of moisture for precipitation events in the ERA-Interim data set (Dee et al., 2011), we obtain the same patterns, but with a substantially shorter time scale of 4-5 days as a global average. We argue that the difference to the 8-10 days that are derived from estimates based on depletion time constants results from several implicit assumptions on how water turnover in the atmosphere can be described. In their discussion paper, van der Ent and Tuinenburg (2016), henceforth VT16, criticize our estimate as incorrect, but do not provide supporting evidence that would indicate that errors have been introduced in our analysis. I reply to some of the statements by VT16 concerning LS16 in this public comment, and suggest several major modifications to their discussion manuscript, which may help to turn it into a more constructive contribution regarding the residence time of atmospheric water vapour.

1. Discrepancies to LS16

a. On pg. 6, L. 11-15, VT16 state:

"All previous estimates referred to in this paper fall within this uncertainty range (Bosilovich and Schubert, 2002; Bosilovich et al., 2002; Chow et al., 1988; van der Ent et al., 2014; Hendriks, 2010; Jones, 1997; Savenije, 2000; UCAR, 2011; Ward and Robinson, 2000; Yoshimura et al., 2004), except for the estimate provided by Läderach and Sodemann (2016) which is less than half, namely 3.9±0.8 days (spatial difference indicated by one standard deviation). Based on the arguments provided above we believe that the latter estimate is incorrect."

The authors claim that the results presented in LS16 are "incorrect" - but no actual evidence is provided that would support that statement. The "arguments provided above" referred to in the quoted paragraph probably relate to an explanation of the estimation of residence times for lakes (see Sec. 3 below). Otherwise the basis of their argument seems to be the reiteration that previous studies have found longer residence times, most of them using the same methods as in VT16. We already were aware of this discrepancy before LS16 was published, and discussed the possible reason for the differences in the paper and the supplement at length. It would be very helpful if the authors could respond to our arguments brought forward in that supplement.

The experiment of Bosilovich et al. (2002) and similar studies also provide a depletion time, rather than a residence time. The same is the case for the WAM method that relates fluxes in a grid cell to the total column water. We do not argue that these calculations are wrong, but that the quantity that is estimated from these methods is not an accurate measure of the residence time as we define it. Bosliovich et al. (2002) by the way even state that "actual residence times should be calculated by taking a Lagrangian approach."

Our method, as all other methods, has uncertainties that we discuss in our manuscript. Because of these uncertainties, we assume that our estimate may be biased low by up to one day, and suggest a range of 4-5 days for the global residence time of moisture. This is far from the range of the residence times expected from depletion time constant calculations. These results are technically correct, and we argue that the disagreement stems mainly from differences between the Lagrangian residence time from evaporation to precipitation as we define it in LS16, and global or local depletion time constants. Of course our diagnostic, as all other methods, is not perfect an relies on some assumptions. Whether the results thus are a "true" residence time will require further work to confirm using other model data sets, and if possible, observations. We argue however that our shorter moisture residence times are in better agreement with global weather system characteristics and thus more plausible than 8-10 days.

b. On pg. 6, L. 29-30, VT16 state:

"Their [Läderach and Sodemann (2016)] spatial patterns are very similar, however, we observe that they underestimate the residence time everywhere with a factor 2–3. This observation leads us to suspect a fundamental irregularity in their method."

In the LS16 paper, we extensively discuss the difference between patterns for the local and global residence time obtained from three different approaches. We restate that our calculations are technically correct, and can be explained in a meaningful manner. To suspect "fundamental irregularities" in the LS2016 method seems far beyond what can be concluded from only using one's own data and methods. A more in-depth analysis would seem appropriate before jumping to such far-reaching conclusions. It may be tempting to suspect calculation errors, but it misses the point. Instead, it may be worthwhile to consider the arguments that we have brought forward.

From a more fundamental perspective, the mere fact that our results are in disagreement with previous results does not allow to conclude on which one is correct or incorrect; it only allows to state disagreement. Consider the (not so small) possibility that because of the many assumptions and data limitations inherent in all methods, it may well be that both estimates are incorrect! As the above two statements in the discussion manuscript are currently written, they represent just a personal opinion, or in the authors' words, a "believe" and "suspicion". In my opinion, such unsubstantiated claims and opinion statements should not be allowed to pass the review process.

It is unfortunate that the authors have not discussed the arguments we brought forward, which do point to the weaknesses and assumptions of depletion time calculations. Admittedly, we only briefly stated this in the main manuscript and fully exemplified them in the electronic supplement due to space limitations. But one would have wished that before submitting their comment the authors would have carefully read the entire publication, including the supplementary material. For ease of discussion, I restate and extend in section 2 below the critical points raised in the supplement to LS16. It would be very valuable if the authors could address these points.

c. On pg. 5, L 1-6, VT16 state:

"The use of Eq. (2) to calculate the global mean residence time of atmospheric water has been criticized by Läderach and Sodemann (2016). They argued that Eq. (2) is not a reliable estimator as it does not involve horizontal moisture transport. Whilst they are correct that location depletion times (van der Ent and Savenije, 2011; Trenberth, 1998) are not equal to actual residence times, we argued above that horizontal moisture transport is irrelevant for the global average value, and that the entire atmospheric volume participates in the hydrological cycle. Thus, Eq. (2) can safely be used to calculate the global average residence time of atmospheric water."

This is not correct. In LS16, we do not argue that horizontal moisture transport is important for Eq. 2. We show that when a depletion time constant approach is used locally, i.e. for individual grid points, it matters substantially for the spatial patterns whether horizontal transport is considered or not (Fig. 2b and c in LS16). The global residence time estimate with or without horizontal moisture fluxes remains however almost the same, as stated in Sec. 4.2 of LS16. This has also been pointed out by reviewer #2 for this discussion paper. In fact, in the supplement text 4 to LS16 we discuss in detail why a difference remains to the estimate of the residence time following Eq. 2 in VT16. We will refer back to this and expand further in the next section.

2. Discussion of simple global mean estimates

A Poission process is a widely-used counting process where events happen at a certain rate, but completely at random. The depletion time constant of the global reservoir of water vapour through precipitation has been used widely to obtain ant estimate of the residence time of atmospheric water vapour. Using a value for global precipitation (which equals global evaporation) of 500 $km^3$/year = 1.37 mm/day and a volume of the global moisture in the atmosphere of 12.7 thousands of $km^3$ (Trenberth et al., 2011), one obtains a global depletion time constant of 12.7 / 1.37 = 9.3 days. Assuming a more extreme case within the range of uncertainty for both quantities, the numbers change to a global precipitation of 616 $km\ yr^{-1}$ = 1.69 $mm\ day^{-1}$ and the global amount of moisture in the atmosphere of 12.3 thousands of $km^3$. This would result in a global average depletion time constant, assumed to be identical to the residence time of moisture, of 12.3 / 1.69 = 7.3 days.

These calculations provide a valuable estimate of how long it takes until the global total column water has been depleted to 1/e by precipitation. But can we interpret this measure as a quantitative proxy for the actual moisture residence time, defined as the time water molecules spend in the atmosphere between evaporation and precipitation? Which assumptions go into considering global precipitation as a Poission process? We present here three arguments against such simple 'back-of-the-envelope' calculations. The first and second are related to the assumptions made when following the arguments of the simple estimate. The third one demonstrates that for systems where the assumptions of a Poission process are violated, depletion time constants do not allow to conclude on the moisture residence time.

a. Precipitation is generated in weather systems of different kind and lifetime. Weather systems are formed, may move through the atmosphere, leading to an unequal distribution of precipitation in space and time, until they decay. This 'intermittent' nature of precipitation is a central aspect, and is also related to the atmospheric residence time of water vapor. Some areas of the world experience frequent and heavy precipitation, other areas, such as deserts, hardly experience any rain. Throughout one year, some areas will thus participate more strongly in the atmospheric water cycle than others. This obvious fact becomes important for the simple estimate when considering that the global mean precipitation of 1.69 mm day assumes that all areas of the earth receive an equal amount of precipitation. According to ERA-Interim reanalyses, 100% of the

global precipitation in a year fall onto 95% of the global surface area, whereas 96% fall onto less than 80% of the surface area (Fig. 1). If we redo the simple estimate from above taking this fact into account, we have to correct the global average rain rates for the actual surface area participating in the water cycle. This would lead us to conclude for example that 90% of the effective global precipitation from the simple number example (1.37/0.8=1.71 mm day and 1.69/0.8=2.11 mm day, respectively) give depletion time constants of 7.2 and 5.8 days, respectively. While somewhat large values may have been selected here, this sensitivity points out that by considering the world's arid areas appropriately results in shorter residence times, in fact already quite a bit closer to the about 4-5 days we obtain from the LS16 method. The simple estimates rely thus on a global uniform distribution of precipitation, and global participation of all surface area, which is in fact not given. In terms of a Poission process, the spatial and temporal coherence of precipitation violates the randomness requirement.

[Figure]

Figure 1: Surface area fraction vs. precipitation fraction from the ERA-Interim reanalysis data (red line). The dashed line would result if precipitation were spatially homogeneous. Reproduced from Läderach and Sodemann (2016).

b. Due to the prevalent atmospheric stratification, different time scales may be relevant at different atmospheric layers. For example, a lower layer of the atmosphere, representing 50% of the column water, or integrated water vapour (IWV), may precipitate and recharge much faster with shallow weather systems. Tropical deep convection may involving the entire column water into precipitation generation, but only exist in some regions such as the ITCZ. A residence time would always consider rain to originate from the entire column, thus neclecting the existence of a faster branch. The assumption that all moisture would be depleted by a "deep" process may contribute to overly large estimates of the moisture residence time from depletion time constants. One may consider that a combination of several Poission processes could represent this complexity in a statistical framework.

c. Consider two hypothetical cases of global temporal precipitation patterns. In the first case, during any given month, rain falls globally every day with an average rain rate of 1.37 mm day[-1]. The same amount of evaporation occurs continously and maintains an atmospheric water volume of $12.7 \times 10^3$ km$^3$. This case will give a depletion time constant of 9.3 days. In a second case, all of the monthly evaporation happens on the first day of each month, and all of the monthly precipitation on the last day. In this second example, the average lifetime of the water vapour is obviously enhanced considerably, while the depletion time constant would still provide the same value of 9.3 days. Obviously, here the stationarity required by a Possion process is not given. Compared to the real atmosphere, both examples are artificial, but they serve to illustrate the point that the depletion time constants do not necessarily faithfully quantify the residence time of

atmospheric water vapour. (This example has been modified from the one given in the Supplement to LS16 to be more to the point of the assumptions underpinning a Possion process).

3. The lake analogy

In the introduction to their Section 3 (pg. 5, L. 24-27), the authors explain some of the reasoning behind a depletion time constant approach:

"Moreover, a lake may be permanently stratified (i.e. there is permanent dead storage) and one could argue that the actual volume participating in the water cycle of the lake does not equal the lake's total volume, meaning that the actual average residence time becomes lower. If one can, however, reliably estimate a lake's volume and in- or outflow, it is not necessary for a lake to be well-mixed for Eq. (2) to hold, the mere necessity is that the entire volume participates in the water cycle. Of course, one could still have significant local differences, but the average can reliably be calculated by Eq. (2)."

The lake analogy serves to illustrate some of the main problems when considering the atmosphere as a hydrological reservoir. For a lake, it may be safe to assume some kind of well-mixed behaviour (or participation) on long time scales. Water vapour is however not well-mixed throughout the atmosphere, most water vapour resides close to the surface, and it travels horizontally over limited distances because of precipitation processes. For the lake, water is the medium, in the atmosphere, air is the medium and water is a trace substance. Following the lake analogy strictly, one should rather compare water vapour in the atmosphere to a tracer that is dissolved in the lake water and has source and removal processes at the surface.

In terms of a Poission process, it may simply be the case that a single random Poisson process does not represent global precipitation adequately. Maybe if one were to use a more realistic representation using several combined Poission processes, or a non-homogeneous Poission process, it may be feasible to obtain a realistic residence time estimate from depletion time constants. While it could be interesting to attempt to represent the atmosphere by a more complex statistical process, we argue that our Lagrangian approach already takes the complexity of the atmosphere into account more realistically that other current approaches.

There are different reservoirs or 'lakes', so to speak, in the atmosphere, some close to the surface that are continuously depleted and replenished by weather systems, and several higher above that only occasionally participate in the atmospheric water cycle, for example during deep convection. The situation varies with latitude and season. If one considers total column water, such as for the global residence time estimate, and as used in the methods of VT16, one implicitly assumes that precipitation extracts water vapour from all atmospheric layers, an assumption that induces large uncertainties in many regions of the world that are dominated by shallower precipitation processes. One consequence of the non-well mixed state of the atmosphere is that one should effectively reduce the IWV in the global average calculation, lowering the residence time. Remaining in the thought framework of a Poission process: If on average 80% of the column water contribute to precipitation processes, IWV would be again be multiplied by a factor of 0.8, resulting in a 1.5-2 day lower residence time in the two examples above, and thus closely approaching the numbers of the Lagrangian residence time estimate of LS16.

A possibly important issue is the question whether some moisture at high elevation resides in the troposphere for very long times, even months. Of course there is very little total water at these elevations, and one can ask the question whether that moisture should be considered for a residence time estimate if it does not

actively take part in the atmospheric water cycle? A meaningful working definition of the residence time of water vapour in the atmosphere could then be more specifically identified as "water vapour in the troposphere that participates in the hydrological cycle on a monthly time scale". Very long-lived water vapour may require different methodological approaches that do not suffer from the accumulation of numerical errors with time.

**4. Simple estimate of the moisture transport distance**

Looking at the problem from another direction, one can ask the question, what are the physical consequences of a moisture residence time of 8-10 days? Fig. 2 depicts the global mean humidity-weighted wind velocity over the entire atmospheric column for September 2005 from ERA-Interim reanalyses. Humidity-weighted wind speeds emphasize lower regions of the atmosphere, where most humidity resides, and gives an indication at what speed most of the humidity in the atmospheric column moves during that month. Values are 10-20 m s$^{-1}$ in the mid-latitudes, lower in the subtropics (4-8 m s$^{-1}$) and 8-12 m s$^{-1}$ at high latitudes. The implication of this is that moisture would travel on average more than 8000 km before precipitating in the extratropics, and more than 4000 km in the subtropics and tropics. Since the 8-10 days is an average value, individual cases will have substantially longer transport distances associated. Considering for example that mid-latitude weather systems develop and intensify, and thereby readily condense large amounts of water vapour along their fronts during 2-3 days, it is difficult to conceive how that corresponds to a 8-10 day time scale and 8000 km length scale of the water transport. In the subtropics, the distance between the evaporation maxima and the ITCZ is only some 15-20 deg in latitude, and also there it is difficult to understand how the moisture can travel for 4000 km on average before precipitating. Values closer to one half of these would be more consistent with expectations from the weather system characteristics in the respective latitudes. While this is not a proof for a shorter residence time, this argument points out that the 8-10 days are not easily explained, even in light of equally simple metrics of moisture lifetime and transport in the atmosphere as the global depletion time estimate.

[Figure]

Figure 2: Humidity weighted horizontal wind velocity (for the entire column, layer by layer) during September 2005 from ERA-Interim reanalyses. Unit is m s$^{-1}$. Range rings around the equator indicate distances of 2000 to 8000 km from the point N0 W0.

5. There are further important points in VT16 that would merit further explanation or discussion:

a. The authors consider 3 variants of the residence time of moisture, termed residence time of precipitation, residence time of evaporation, and age of water vapour. No explanation is given on how the age has been calculated. I assume all of these are different projections of the same quantity (precipitation RT projects forward, evaporation RT projects backward), and should have the same mean value. The relation and difference between each way of presenting the residence time could be stated more clearly to avoid confusion of the readers.

b. The Tuinenburg method is based on the Dirmeyer and Brubaker (1999) approach (a corresponding reference is missing in the manuscript). As I understand that method, several isentropic trajectories are calculated from every 0.5x0.5deg grid point at several elevations, then surface evaporation is accumulated along these trajectories at every time step. Essentially, that method thereby assumes a well-mixed atmosphere at every time step and grid point - because the vertical position of the trajectory does not matter. Moreover, water vapour is assumed to be a conserved quantity once it is mixed into the air parcel (i.e. precipitation does not remove earlier moisture contributions). The method is furthermore sensitive to the reliability of the evaporation data set. This method clearly relies on strong assumptions, in particular compared to our Lagrangian method (Sodemann et al., 2008) which was applied in LS16 and neither assumes well-mixed conditions nor relies on evaporation, which is a difficult variable to observe and has large local uncertainties, in particular when derived from satellite observations (Rodell et al., 2015).

Interestingly however, the median of the residence time with VT16's Lagrangian method 3D-T are with 5.7 and 4.6 days clearly lower than 8-10 days. There is a lot of very short-lived water vapour identified by this method. VT16 state that the very long tail leads to a mean to 8-10 days. Taken at face value, the low median argues for a residence time of the bulk of the water vapour of much less than 8-10 days. With trajectory calculation times exceeding 10-15 days, the tail gets more and more uncertain, in particular if few trajectories per grid point are considered. What would be the residence time if evaluation was not cut off at 30 (pg. 5 L. 10), but at 20 or, say, 50 days? Would it still be possible to argue that the mean is representative of the distribution? One consequence of this difference between the mean and the median is that the results in Figure 2 of VT16 should be shown separately for the WAM and the 3D-T method.

c. A particularly puzzling result is shown in Fig. 3 of VT16. The patterns of the seasonal residence time appear difficult to interpret physically. Residence times increase from about 5 days to more than 15 days during northern hemisphere winter over the North Pacific, and the reverse applies to the southern hemisphere. In the current version, VT16 do not provide further explanation on what could cause this result, and how it relates to observed seasonal changes in the climate system. It would be interesting to learn about a corresponding strong change in the climate system that would explain such a drastic seasonal change.

**References**

Dee, D. P., Uppala, S. M., Simmons, A. J., Berrisford, P., Poli, P., Kobayashi, S., Andrae, U., Balmaseda, M. A., Balsamo, G., Bauer, P., Bechtold, P., Beljaars, A. C. M., van de Berg, L., Bidlot, J., Bormann, N., Delsol, C., Dragani, R., Fuentes, M., Geer, A. J., Haimberger, L., Healy, S. B., Hersbach, H., Hólm, E. V., Isaksen, L., Kållberg, P., Köhler, M., Matricardi, M., McNally, A. P., Monge-Sanz, B. M., Mor- crette, J. J., Park, B. K., Peubey, C., de Rosnay, P., Tavolato, C., Thépaut, J. N., and Vitart, F.: The ERA-Interim reanalysis: Configuration and performance of the data assimilation system, Q. J. R. Meteorol. Soc., 137, 553–597, doi:10.1002/qj.828, 2011.

Dirmeyer, P. A., and K. L. Brubaker (1999), Contrasting evaporative moisture sources during the drought of 1988 and the flood of 1993, J. Geophys. Res., 104(D16), 19,383–19,397.

Läderach, A. and Sodemann, H.: A revised picture of the atmospheric moisture residence time, Geophys. Res. Lett., 43, 924–933, doi:10.1002/2015GL067449, 2016.

Rodell, M., Beaudoing, H. K., L'Ecuyer, T. S., Olson, W. S., Famiglietti, J. S., Houser, P. R., Adler, R., Bosilovich, M. G., Clayson, C. A., Chambers, D., Clark, E., Fetzer, E. J., Gao, X., Gu, G., Hilburn, K., Huffman, G. J., Lettenmaier, D. P., Liu, W. T., Robertson, F. R., Schlosser, C. A., Sheffield, J., and Wood, E. F.: The observed state of the water cycle in the early twenty-first century, J. Clim., 28, 8289–8318, doi:10.1175/JCLI-D-14-00555.1, 2015.

Sodemann, H., C. Schwierz, and H. Wernli (2008), Interannual variability of Greenland winter precipitation sources: Lagrangian moisture diagnostic and North Atlantic Oscillation influence, J. Geophys. Res., 113, D03107, doi: 10.1029/2007JD008503.

van der Ent, R. and Tuinenburg, O.: The residence time of water in the atmosphere revisited, Hydrol. Earth Syst. Sci. Discuss., doi:10.5194/hess-2016-431, 2016.

---

## Author Comment (AC5) · 19 Oct 2016

**The residence time of water in the atmosphere revisited**

Ruud J. van der Ent[1] and Obbe A. Tuinenburg[2]

[1]Department of Physical Geography, Faculty of Geosciences, Utrecht University, Utrecht, the Netherlands
[2]Department of Environmental Sciences, Copernicus Institute for Sustainable development, Utrecht University, Utrecht, the Netherlands

*Correspondence to:* Ruud J. van der Ent (r.j.vanderent@uu.nl)

**Response to Anonymous Referee #3**

We thank Anonymous Referee #3 for reviewing our manuscript. It is a pity, however, that the comments consist of the referee's "feelings" and general statements without discussing any specific parts of our manuscript or stating anything concrete. Comments by the referee are in italic and replies are in normal text. The detailed adjustments to the revised manuscript will follow

5    after the public discussion period.

**Proof that the global average residence time of water in the atmosphere is 8–10 days**

First, we would like to respond to the title of comments provided by Anonymous Referee #3, in which he or she claims that our paper does not provide a counterargument for the study of Läderach and Sodemann (2016), who claimed the global average

10   residence time to be be 4–5 days (more precisely 3.9±0.8 days for 15-day backward trajectories). It is true that we do no provide a counter argument in the sense that we identify a flaw in their method, however, our counter argument consists of the finding that their outcome is inconsistent with the state-of-the-art data of the hydrological cycle, including the dataset they use themselves. In Section 3 of our paper we clearly explain how one can derive the global average residence time of water in atmosphere from relative simple mass balance calculations of well-established estimations of the stocks and fluxes in the

15   global hydrological cycle. The following text (cited from Section 3) disproofs the estimate of Läderach and Sodemann (2016):

"If one would like to know the average residence time $\tau$ in any reservoir one simply divides its average mass $\overline{M}$ (or volume when assuming constant density) by its average outgoing mass flux $\overline{F}$ (which equals the ingoing flux when there is no change of mass):

20   $$\tau = \frac{\overline{M}}{\overline{F}}. \tag{2}$$

Whereas this is a simple formula, computation of reliable residence times in, for example, a surface water lake may be difficult due to many uncertainties in a lake's volume, hydraulic flow, precipitation, evaporation and seepage (Monsen et al., 2002). Moreover, a lake may be permanently stratified (i.e. there is permanent dead storage) and one could argue that the actual volume participating in the water cycle of the lake does not equal the lake's total volume, meaning that the actual average

25   residence time becomes lower. If one can, however, reliably estimate a lake's volume and in- or outflow, it is not necessary

for a lake to be well-mixed for Eq. (2) to hold, the mere necessity is that the entire volume participates in the water cycle. Of course, one could still have significant local differences, but the average can reliably be calculated by Eq. (2).

When the Earth's entire atmosphere is considered to be the reservoir of study, its residence time can actually be calculated much easier than that of a lake. In the global case the only inflow is evaporation and the only outflow is precipitation. Moreover, due to the turbulent nature of the atmosphere all water that resides in the atmosphere also participates in the atmospheric water cycle, i.e., there is no such thing as permanent dead water storage (e.g., Jacobson, 1999).

The use of Eq. (2) to calculate the global mean residence time of atmospheric water has been criticized by Läderach and Sodemann (2016). They argued that Eq. (2) is not a reliable estimator as it does not involve horizontal moisture transport. Whilst they are correct that location depletion times (van der Ent and Savenije, 2011; Trenberth, 1998) are not equal to actual residence times, we argued above that horizontal moisture transport is irrelevant for the global average value, and that the entire atmospheric volume participates in the hydrological cycle. Thus, Eq. (2) can safely be used to calculate the global average residence time of atmospheric water.

Applying Eq. (2) on estimates of the global hydrological cycle (Fig. 1) yields a global mean residence time of atmospheric water of 8.9±0.4 days (uncertainty indicated by one standard deviation). The calculation of the mean is as follows:

$$\tau = \frac{12.6 \cdot 10^3}{(403.5 + 116.5) \cdot 10^3} = 0.024 \text{ years} = 8.9 \text{ days,} \tag{3}$$

and the standard deviation was calculated by general uncertainty propagation theory. The $1^{\text{th}}$ and $99^{\text{th}}$ percentile of this estimate are 7.9 and 9.8 days respectively. All previous estimates referred to in this paper fall within this uncertainty range (Bosilovich and Schubert, 2002; Bosilovich et al., 2002; Chow et al., 1988; van der Ent et al., 2014; Hendriks, 2010; Jones, 1997; Savenije, 2000; UCAR, 2011; Ward and Robinson, 2000; Yoshimura et al., 2004), except for the estimate provided by Läderach and Sodemann (2016) which is less than half, namely 3.9±0.8 days (spatial difference indicated by one standard deviation). Based on the arguments provided above we believe that the latter estimate is incorrect."

**Novel contributions**

As one can read above, our proof for the residence time to be 8–10 days is not necessarily based on our own tracking methods (WAM-2layers and 3D-Trajectories). The results obtained with FLEXPART by Läderach and Sodemann do not make sense from a mass balance point of view (Eqs. (2) and (3)). If *Geophysical Research Letters*, wherein the work of Läderach and Sodemann (2016) was published, did not have the policy not to accept comments on published papers we could have submitted the text above.

In contrast to the work of Läderach and Sodemann (2016) our tracking methods yield trustworthy estimates of residence time in the range which is to be expected. However, this paper goes beyond being a commentary on another paper as our novel contributions include:

– Extension of the standard global hydrological cycle picture to show not only the fluxes, but also the atmospheric residence time associated with these fluxes, split out over land and ocean;

- Spatial maps of global residence time of atmospheric water. This study is the first to look at this from both the evaporation and precipitation perspective. Moreover, interesting summer vs. winter patterns are revealed for the first time;

- Global probability density functions of residence time of atmospheric water, from a land and ocean perspective, are shown for the first time.

**Response to subsequent comments of Anonymous Referee #3**

*This paper looks to address the question regarding the residence time of water in the atmosphere recently raised by Läderach and Sodemann (2016). I feel this study does not provide enough evidence to significantly contribute to the discussion.*

Indeed we address the question of residence time of water in the atmosphere, but the issue of atmospheric residence time was raised by many others before. Our paper revisits this literature and demonstrates with different methods – global water balance and two tracking models – that the global average residence time is 8–10 days, and thus, that the main result of Läderach and Sodemann (2016) – suggesting it to be 4–5 days – is erroneous. We provide the main evidence in Section 3, which we even called "Why the global average residence time of water in the atmosphere is 8–10 days".

*1) There are large limitations to using tracking models to track atmospheric moisture. The authors need to go into a lot more detail as to how the models were applied to this dataset. Specifically, how the water in the model is tagged, how this tagged water relates to the evaporation and precipitation in the subsequent time step, whether mixing of this water is taken into account. Without going into more detail as to how the tracking models specifically deal with water it is not possible to address question of the residence time of the water.*

The tracking methods we apply here – WAM-2layers and 3D-Trajectories – have been tested and validated (see van der Ent et al., 2013). There are obviously some assumptions in the methods applied, but by applying both we also test the sensitivity of the outcome to the tracking model used. We conclude that there is some sensitivity, but both methods yield global average numbers which make sense when compared to the global water balance (see Fig. 1). This cannot be said from the results produced by Läderach and Sodemann (2016), which use a single method (FLEXPART) and yields too low estimates that cannot physically be true if one looks at the numbers of the global water balance, including the estimates from their own input data (ERA-Interim). The devil is not in the details, as the referee suggests, but whether the results match with the fundamental mass balance equation (see Eqs. (2) and (3)), or not, and ours do.

*2) The authors highlight the difference in the assumptions made by Laderach and Sodermann (2016), but more discussion as to why the authors disagree with these differences is needed. No evidence is provided to support the authors' choice in assumptions over the choice by Laderach and Sodermann.*

Actually, we do not highlight the difference in the assumptions made, where did the referee read this? The only thing we can and did say is that the length of their trajectories is is too short, namely 15 or 20 days, while we find that about 5 % of the atmospheric moisture has a residence time of more than 30 days (Fig. 4). In the revised manuscript we will highlight this more in Section 5, which deals with the probability density functions of atmospheric residence time. However, this is by far not

enough to explain the factor 2 difference. As we both find very similar spatial patterns (compare our Fig 2c to Läderach and Sodemann, 2016, Fig. 2a), and we both use ERA-Interim data, we can only conclude that there is a fundamental irregularity in their method. For all we know they could have accidentally divided all their results in the post-processing by a factor 2 using a wrong time step, flux or stock. However, this is wild speculation, and it is not within our possibilities nor our task to get to the

5    bottom of this out within the scope is this paper. The only thing we can and did do is highlighting the difference in obtained results and showing that theirs is unphysical (see Section 3).

*3) The analysis of the results conclude that ERA-Interim shows close agreement to the previous studies of residence time, however does not provide a response to Laderach and Sodermann.*

10    In Section 3 we show that using data from the state-of-the-art estimates of the global hydrological cycle (Rodell et al., 2015; Trenberth et al., 2011) yields a global mean residence time of atmospheric water of $8.9\pm0.4\,\mathrm{days}$ (uncertainty indicated by one standard deviation), with the 1th and 99th percentile of this estimate being 7.9 and $9.8\,\mathrm{days}$ respectively. That is the response and proof that the estimates of Läderach and Sodemann (2016), namely $4$–$5\,\mathrm{days}$ or $3.9\pm0.8\,\mathrm{days}$ (spatial variability indicated by one standard deviation) for 15-day backward trajectories or $4.4\,\mathrm{days}$ for 20-day backward trajectories, are so far outside the

15    uncertainty range that they cannot be true.

*The paper needs to go into more detail of the tracking methods, a more detailed analysis of the results and a greater discussion regarding the assumptions made by Laderach and Sodermann, and to provide evidence as to why the authors disagree with these assumptions.*

20    As explained above, we believe that we have already made a very strong case for the results of Läderach and Sodemann (2016) to be wrong. Please note that we mean their absolute values, we do not doubt their spatial patterns, which we actually found innovative. As mentioned also in our response to Kevin Trenberth we will add a paragraph to the Supplement that discusses the differences between our own methods WAM-2layers and 3D-T (Fig. S1) in terms of their underlying assumptions. However, it is beyond our possibilities nor within scope of this paper to explore what went exactly wrong in the analysis of

25    Läderach and Sodemann (2016). The latter are, in our opinion, the only ones that could figure that out. Rather than focusing on the tracking methods, however, if the referee or anyone else disagrees with our estimates we would like to dare them to challenge Eqs. (2) and (3).

**References**

30    Bosilovich, M. G. and Schubert, S. D.: Water vapor tracers as diagnostics of the regional hydrologic cycle, J. Hydrometeorol., 3(2), 149–165, doi:10.1175/1525-7541(2002)003<0149:WVTADO>2.0.CO;2, 2002.

Bosilovich, M. G., Sud, Y., Schubert, S. D. and Walker, G. K.: GEWEX CSE sources of precipitation using GCM water vapor tracers, GEWEX News, 12(3), 1,6–7,12, 2002.

Chow, V. T., Maidment, D. R. and Mays, L. W.: Applied Hydrology, McGraw-Hill, Singapore., 1988.

35    Hendriks, M. R.: Introduction to Physical Hydrology, Oxford University Press, New York., 2010.

Jones, J. A. A.: Global Hydrology: processes, resources and environmental management, Longman, Harlow., 1997.

Läderach, A. and Sodemann, H.: A revised picture of the atmospheric moisture residence time, Geophys. Res. Lett., 43, 924–933, doi:10.1002/2015GL067449, 2016.

Monsen, N. E., Cloern, J. E., Lucas, L. V. and Monismith, S. G.: A comment on the use of flushing time, residence time, and age as transport time scales, Limnol. Oceanogr., 47(5), 1545–1553, doi:10.4319/lo.2002.47.5.1545, 2002.

Rodell, M., Beaudoing, H. K., L'Ecuyer, T. S., Olson, W. S., Famiglietti, J. S., Houser, P. R., Adler, R., Bosilovich, M. G., Clayson, C. A., Chambers, D., Clark, E., Fetzer, E. J., Gao, X., Gu, G., Hilburn, K., Huffman, G. J., Lettenmaier, D. P., Liu, W. T., Robertson, F. R., Schlosser, C. A., Sheffield, J. and Wood, E. F.: The observed state of the water cycle in the early twenty-first century, J. Clim., 28(21), 8289–8318, doi:10.1175/JCLI-D-14-00555.1, 2015.

Savenije, H. H. G.: Water scarcity indicators; the deception of the numbers, Phys. Chem. Earth Part B Hydrol. Ocean. Atmos., 25(3), 199–204, doi:10.1016/S1464-1909(00)00004-6, 2000.

Trenberth, K. E.: Atmospheric moisture residence times and cycling: Implications for rainfall rates and climate change, Clim. Change, 39(4), 667–694, doi:10.1023/A:1005319109110, 1998.

Trenberth, K. E., Fasullo, J. T. and Mackaro, J.: Atmospheric Moisture Transports from Ocean to Land and Global Energy Flows in Reanalyses, J. Clim., 24(18), 4907–4924, doi:10.1175/2011jcli4171.1, 2011.

UCAR: The Water Cycle, [online] Available from: http://scied.ucar.edu/longcontent/water-cycle (Accessed 22 July 2016), 2011.

van der Ent, R. J. and Savenije, H. H. G.: Length and time scales of atmospheric moisture recycling, Atmos. Chem. Phys., 11(5), 1853–1863, doi:10.5194/acp-11-1853-2011, 2011.

van der Ent, R. J., Wang-Erlandsson, L., Keys, P. W. and Savenije, H. H. G.: Contrasting roles of interception and transpiration in the hydrological cycle - Part 2: Moisture recycling, Earth Syst. Dyn., 5(2), 471–489, doi:10.5194/esd-5-471-2014, 2014.

Ward, R. C. and Robinson, M.: Principles of Hydrology, McGraw-Hill, Berkshire., 2000.

Yoshimura, K., Oki, T., Ohte, N. and Kanae, S.: Colored moisture analysis estimates of variations in 1998 Asian monsoon water sources, J. Meteorol. Soc. Japan, 82(5), 1315–1329, doi:10.2151/jmsj.2004.1315, 2004.

[Figure]

**Figure 1.** Earth's hydrological cycle with residence times (2002–2008). All residence times shown are weighted averages. The uncertainty ranges indicated for the residence times from the moisture tracking methods refer to the uncertainty associated with model choice (WAM-2layers or 3D-Trajectories). Tre11 stands for Trenberth et al. (2011) and Rod15 stands for Rodell et al. (2015). The land area is $147 \cdot 10^6 \, \text{km}^2$ and the ocean area is $363 \cdot 10^6 \, \text{km}^2$.

---

## Author Comment (AC6) · 15 Nov 2016

**The residence time of water in the atmosphere revisited**

Ruud J. van der Ent[1] and Obbe A. Tuinenburg[2]

[1]Department of Physical Geography, Faculty of Geosciences, Utrecht University, Utrecht, the Netherlands
[2]Department of Environmental Sciences, Copernicus Institute for Sustainable development, Utrecht University, Utrecht, the Netherlands

*Correspondence to:* Ruud J. van der Ent (r.j.vanderent@uu.nl)

**Response to Harald Sodemann**

We would to thank Harald Sodemann for his extensive comment concerning the differences between the results of Läderach and Sodemann (2016), henceforth LS16, and our paper, henceforth VT16. Despite the fact the we disagree with almost all points raised in his comment we appreciate the constructive attitude that was expressed in the comment as well as during personal discussions we had during the 8[th] EGU Leonardo Conference (25–27 October, Ourense, Spain). Below we discuss all the point raised by Harald Sodemann in the sequence of the posted comment. Points raised by Harald Sodemann are in italic and our replies are in normal text. The detailed adjustments to the revised manuscript will follow after the public discussion period. Due to the length of this discussion we would like to point out that the most fundamental issues in our opinion are discussed in Section 2 of this response.

**Introduction**

*Until recently, it was commonly accepted knowledge that the residence time of water vapor should be about 8–10 days on a global average. In our recent study published in Geophysical Research Letters (Läderach and Sodemann, 2016, henceforth LS16) we challenged this viewpoint with regard to different aspects. First, we pointed out deficiencies in the use of depletion time constants if used as a local measure of the time water vapour resides in the atmosphere between evaporation and precipitation. From a modified Eulerian estimate taking into account horizontal moisture flux between grid cells, we derive more plausible patterns that are in correspondence with prevailing weather systems. Using a Lagrangian method that identifies the sources of moisture for precipitation events in the ERA-Interim data set (Dee et al., 2011), we obtain the same patterns, but with a substantially shorter time scale of 4–5 days as a global average.*

We also state in our manuscript that LS16 are correct that location depletion times (van der Ent and Savenije, 2011; Trenberth, 1998) are not equal to actual residence times. Van der Ent and Savenije (2011) stated that their calculated metrics are to be interpreted as local timescales for precipitation and evaporation, which give an indication for the residence time of atmospheric moisture if horizontal moisture transport is small. As their stated objective was to calculate local metrics there was, however, in our opinion never any controversy about the fact that these could not be interpreted as actual residence times.

Besides the Lagrangian method used by LS16, from which they obtain a 4–5 days residence time estimate, it is interesting to note that LS16 also use a local Eulerian method including a transport approximation and find a global annual mean value of the time 'constant' to be 8.3±4.2 days (spatial variability indicated by 1 standard deviation). While LS16 disregard this Eulerian estimate as incorrect, because "precipitation processes are strongly oversimplified", we interpret this result as supporting evidence that the 8–10 days must be correct due to the most elemental physical concept that an average residence time can be calculated from dividing a stock by its average influx or outflux under the assumption that there is negligible net stock change over a longer period, and that this applies to the atmosphere as well (e.g., Chow et al., 1988; Hendriks, 2010; Jones, 1997;

Savenije, 2000; UCAR, 2011; Ward and Robinson, 2000).

*We argue that the difference to the 8–10 days that are derived from estimates based on depletion time constants results from several implicit assumptions on how water turnover in the atmosphere can be described. In their discussion paper, van der Ent and Tuinenburg (2016), henceforth VT16, criticize our estimate as incorrect, but do not provide supporting evidence that*

*would indicate that errors have been introduced in our analysis. I reply to some of the statements by VT16 concerning LS16 in this public comment, and suggest several major modifications to their discussion manuscript, which may help to turn it into a more constructive contribution regarding the residence time of atmospheric water vapour.*

Sodemann uses the word "constants" often in his arguments, but we will argue below that we do in fact not make any assumption of constant fluxes. Assuming that global estimates of evaporation, precipitation and atmospheric storage are correct (within a certain uncertainty range, see VT16 Fig. 1), we will argue that deviations from a constant flux can only cause changes in the probability density function of global average residence time, but not in the actual average, as that would violate mass balance.

We purposely chose not to discuss the possible issues in the applied methodology of LS16 that lead to an underestimation of the global average residence time, as we thought that VT16 Section 3 provides enough proof of the 4–5 day estimate to be incor- rect, and then it is up to Läderach and Sodemann to find out how they can improve their methodology. As Sodemann is not convinced we will provide some more detailed criticism of their applied methodology. Our latest insights into the reason for the underestimation of the global average residence time by LS16 come from Stohl and James (2004), who evaluated the FLEXPART methodology. FLEXPART is generally only used to look at specific humidity changes (evaporation – precipitation, or $E - P$), but they found that when FLEXPART is used to evaluate $E$ and $P$ separately, globally averaged, $E = 1380\,\mathrm{mm\,yr^{-1}}$, which corresponds to $E = 704 \cdot 10^3\,\mathrm{km^3\,yr^{-1}}$. Since LS16 track precipitation backward with FLEXPART they essentially suffer from this likely overestimation of evaporation, and hence the underestimation of the residence time. LS16 did not evaluate how much evaporation they attributed in total in their runs, however, using the numbers obtained by Stohl and James (2004) and ERA-Interim atmospheric storage, a global average residence time of $12.4 \cdot 10^3\,\mathrm{km^3}/704 \cdot 10^3\,\mathrm{km^3\,yr^{-1}} = 0.017\,\mathrm{years} = 6.4\,\mathrm{days}$ is obtained. The assumption that the applied methodology can accurately attribute evaporation and estimate precipitation, thus has likely major influence on the results by LS16. Moreover, Sodemann et al. (2008) themselves note that a number of moisture transport processes is neglected, which are moisture changes due to convection, turbulence, numerical diffusion, and rainwater evaporation. Stohl and Seibert (1998) even note that specific humidity fluctuations along a trajectory may be entirely unphysical. In contrast to FLEXPART – WAM-2layers and 3D-Trajectories – do not suffer from this issue as evaporation and precipitation fields are used explicitly.

**1. Discrepancies to LS16**

*a. On pg. 6, L. 11-15, VT16 state: "All previous estimates referred to in this paper fall within this uncertainty range (Bosilovich and Schubert, 2002; Bosilovich et al., 2002; Chow et al., 1988; van der Ent et al., 2014; Hendriks, 2010; Jones, 1997; Savenije, 2000; UCAR, 2011; Ward and Robinson, 2000; Yoshimura et al., 2004), except for the estimate provided by Läderach and Sodemann (2016) which is less than half, namely 3.9±0.8 days (spatial difference indicated by one standard deviation). Based on the arguments provided above we believe that the latter estimate is incorrect."*

The authors claim that the results presented in LS16 are "incorrect" - but no actual evidence is provided that would support that statement. The "arguments provided above" referred to in the quoted paragraph probably relate to an explanation of the estimation of residence times for lakes (see Sec. 3 below). Otherwise the basis of their argument seems to be the reiteration that previous studies have found longer residence times, most of them using the same methods as in VT16. We already were aware of this discrepancy before LS16 was published, and discussed the possible reason for the differences in the paper and the supplement at length. It would be very helpful if the authors could respond to our arguments brought forward in that supplement.

The basis for our arguments evolves around VT16 Eqs. (2) and (3). Based on that, the likelihood of the global average residence time of atmospheric moisture to be lower than $3.9 \, \mathrm{days}$ corresponds to $1 \cdot 10^{-30}$. This is so highly improbable that in non-statistical terms this may be translated into incorrect. We simply note that the estimates of all other studies do fall within the $1^{\text{th}}$ and $99^{\text{th}}$ percentile of this estimate (7.9 and $9.8 \, \mathrm{days}$ respectively). In the revised version we will write more explicitly that Eqs. (2) and (3) for the basis for our argument. These equations, together with the notion that all atmospheric water sooner or later participates in the hydrological cycle (i.e., no dead storage), also implicitly falsifies the arguments that were brought forward in the supplement of LS16. These arguments by LS16 are repeated below and we will argue that these are not valid.

*The experiment of Bosilovich et al. (2002) and similar studies also provide a depletion time, rather than a residence time. The same is the case for the WAM method that relates fluxes in a grid cell to the total column water. We do not argue that these calculations are wrong, but that the quantity that is estimated from these methods is not an accurate measure of the residence time as we define it. Bosilovich et al. (2002) by the way even state that "actual residence times should be calculated by taking a Lagrangian approach."*

In WAM-2layers we use water tagging; in the forward(backward) scheme water is tagged as it evaporates(precipitates) from a certain masked area, transported through the atmosphere in two layers and at each time step the ratio of tagged to total water is calculated as well as the age of this water (VT16, Eq. (1)). The assumption is indeed that precipitation stems from the total column water. Surely this is not likely to be true everywhere, but performs generally well in monsoon regions (Fitzmaurice, 2007). In any case, this cannot be seen as a counterargument for the validity of the 8–10 day estimate. If certain precipitation events are not "well-mixed" but stem only from let's say the bottom $75\,\%$ of the atmospheric water then this would only influence the probability density function (PDF) of precipitation residence time: $75\,\%$ of the particles have lower residence times, but other $25\,\%$ of the particles reside even longer in the atmosphere contributing to even longer tails than the ones we found in VT16 Fig. 4.

Depletion times could be calculated with WAM-2layers if all water was tagged at time step $t_0$, without renewal of tagged water, but this is not the experiment that we performed. Bosilovich et al. (2002) performed exactly such an experiment and found this to be $9.2\,\text{days}$. They write "... indicates an average global residence time of $9.2\,\text{days}$ from this simulation." Strictly speaking Sodemann is correct that they calculated a global average depletion time, but there is no physical reason why – globally averaged – depletion time should be different from residence time. The quote that "actual residence times should be calculated by taking a Lagrangian approach." is actually from van der Ent and Savenije (2011) whereby they refer to the Semi-Lagrangian scheme of Bosilovich et al. (2002), but this quote is outdated as van der Ent et al. (2014) have shown that an Eulerian moisture tracking method is also capable of calculating residence times.

*Our method, as all other methods, has uncertainties that we discuss in our manuscript. Because of these uncertainties, we assume that our estimate may be biased low by up to one day, and suggest a range of 4–5 days for the global residence time of moisture. This is far from the range of the residence times expected from depletion time constant calculations. These results are technically correct, and we argue that the disagreement stems mainly from differences between the Lagrangian residence time from evaporation to precipitation as we define it in LS16, and global or local depletion time constants. Of course our diagnostic, as all other methods, is not perfect and relies on some assumptions. Whether the results thus are a "true" residence time will require further work to confirm using other model data sets, and if possible, observations. We argue however that our shorter moisture residence times are in better agreement with global weather system characteristics and thus more plausible than 8–10 days.*

See our reply under the introduction of this response. In addition, the length of the trajectories of 15 or $20\,\text{days}$ used by LS16 is insufficient to calculate the global average residence time reliably (see VT16, Fig. 4). The value of $8.9\pm0.4\,\text{days}$ (uncertainty indicated by one standard deviation) that we present in VT16 is a best estimate given our current knowledge of the global hydrological cycle.

*b. On pg. 6, L. 29-30, VT16 state: "Their [Läderach and Sodemann (2016)] spatial patterns are very similar, however, we observe that they underestimate the residence time everywhere with a factor 2–3. This observation leads us to suspect a fundamental irregularity in their method."*

*In the LS16 paper, we extensively discuss the difference between patterns for the local and global residence time obtained from three different approaches. We restate that our calculations are technically correct, and can be explained in a meaningful manner. To suspect "fundamental irregularities" in the LS2016 method seems far beyond what can be concluded from only using one's own data and methods. A more in-depth analysis would seem appropriate before jumping to such far-reaching conclusions. It may be tempting to suspect calculation errors, but it misses the point. Instead, it may be worthwhile to consider the arguments that we have brought forward.*

We will remove this speculative statement about "fundamental irregularities". In the introduction, however, we will add the notion that the methodology by LS16 suffers from the assumption that the applied methodology can accurately attribute evaporation and estimate precipitation, likely overestimates evaporation (Stohl and James, 2004), and that this has not been validated by LS16.

*From a more fundamental perspective, the mere fact that our results are in disagreement with previous results does not allow to conclude on which one is correct or incorrect; it only allows to state disagreement. Consider the (not so small) possibility that because of the many assumptions and data limitations inherent in all methods, it may well be that both estimates are incorrect! As the above two statements in the discussion manuscript are currently written, they represent just a personal opinion, or in the authors' words, a "believe" and "suspicion". In my opinion, such unsubstantiated claims and opinion statements should not be allowed to pass the review process.*

As explained in Section 1a of this response, the conclusion that the results of LS16 are incorrect does not stem from mere comparison with previous research.

*It is unfortunate that the authors have not discussed the arguments we brought forward, which do point to the weaknesses and assumptions of depletion time calculations. Admittedly, we only briefly stated this in the main manuscript and fully exemplified them in the electronic supplement due to space limitations. But one would have wished that before submitting their comment the authors would have carefully read the entire publication, including the supplementary material. For ease of discussion, I restate and extend in section 2 below the critical points raised in the supplement to LS16. It would be very valuable if the authors could address these points.*

VT16 Eqs. (2) and (3), together with the notion that all atmospheric water sooner or later participates in the hydrological cycle (i.e., no dead storage), also implicitly falsifies the arguments that were brought forward in the supplement of LS16. These arguments by LS16 are repeated below and we will argue that these are not valid.

*c. On pg. 5, L 1-6, VT16 state: "The use of Eq. (2) to calculate the global mean residence time of atmospheric water has been criticized by Läderach and Sodemann (2016). They argued that Eq. (2) is not a reliable estimator as it does not involve horizontal moisture transport. Whilst they are correct that location depletion times (van der Ent and Savenije, 2011; Trenberth, 1998) are not equal to actual residence times, we argued above that horizontal moisture transport is irrelevant for the global average value, and that the entire atmospheric volume participates in the hydrological cycle. Thus, Eq. (2) can safely be used to calculate the global average residence time of atmospheric water."*

This is not correct. In LS16, we do not argue that horizontal moisture transport is important for Eq. 2. We show that when a depletion time constant approach is used locally, i.e. for individual grid points, it matters substantially for the spatial patterns whether horizontal transport is considered or not (Figs. 2b and c in LS16). The global residence time estimate with or without horizontal moisture fluxes remains, however, almost the same, as stated in Sec. 4.2 of LS16. This has also been pointed out by reviewer #2 for this discussion paper. In fact, in the supplement text 4 to LS16 we discuss in detail why a difference remains to the estimate of the residence time following Eq. 2 in VT16. We will refer back to this and expand further in the next section.

We will add a Section to our own Supplement were we answer the questions raised by LS16 (Supplement Section 4). The same points also appear in Section 2 below.

We intend to change this sentence into: "The use of Eq. (2) to calculate the global mean residence time of atmospheric water has been criticized by Läderach and Sodemann (2016). They argued that Eq. (2) is a) not a reliable estimator for local residence times as it does not involve horizontal moisture transport, b) should be corrected for the surface are of the Earth where most precipitation is observed, and c) that the temporal characteristics of global precipitation cannot measured by depletion time constants. However, a*) horizontal moisture transport is irrelevant for the global average value, b*) the surface area of the Earth is irrelevant as it is not in Eq. (2), but nonetheless all areas participate in the hydrological cycle as there is also transport even over the Sahara (e.g., Schicker et al., 2010; Goessling and Reick, 2013), and c*) the values in Eq. (2) correspond to the elemental physical concept that an average residence time can be calculated from dividing a stock by its average influx or outflux under the assumption that there is negligible net stock change over a longer period; whether these average fluxes are constant or not is irrelevant, and the temporal characteristics of precipitation and evaporation can only affect the PDFs of the residence time but not the average. In the Supplement we show that the counterexamples objecting to Eq. (2) by Läderach and Sodemann (2016, Supplement Section 4) can easily be falsified. Moreover, we argued above that the entire atmospheric volume participates in the hydrological cycle. Thus, Eq. (2) can safely be used to calculate the global average residence time of atmospheric water."

**2. Discussion of simple global mean estimates**

*A Poission process is a widely-used counting process where events happen at a certain rate, but completely at random. The depletion time constant of the global reservoir of water vapour through precipitation has been used widely to obtain ant estimate of the residence time of atmospheric water vapour. Using a value for global precipitation (which equals global*

*evaporation) of 500 $\mathrm{km}^3$/year = 1.37 mm/day and a volume of the global moisture in the atmosphere of 12.7 thousands of $\mathrm{km}^3$ (Trenberth et al., 2011), one obtains a global depletion time constant of $12.7/1.37 = 9.3$ days. Assuming a more extreme case within the range of uncertainty for both quantities, the numbers change to a global precipitation of $616 \mathrm{kmyr}^{-1} = 1.69 \mathrm{mm}\, \mathrm{day}^{-1}$ and the global amount of moisture in the atmosphere of 12.3 thousands of $\mathrm{km}^3$. This would result in a global average depletion time constant, assumed to be identical to the residence time of moisture, of $12.3/1.69 = 7.3$ days.*

First of all, Sodemann is mixing up units here. The likely value for the precipitation/evaporation flux is a factor 1000 off and should be $500 \cdot 10^3\, \mathrm{km}^3/\mathrm{yr}$. Divided by $365.25$ days this equals $1.37 \cdot 10^3\, \mathrm{km}^3\, \mathrm{day}^{-1}$. Averaged over the Earth's surface area $(510 \cdot 10^6\, \mathrm{km}^2)$, this equals $2.68\, \mathrm{mm}\, \mathrm{day}^{-1}$. The calculated corresponding residence times where, however, computed correctly as 9.3 and 7.3 days, respectively for a normal and extreme case. It should als be noted that the extreme case is indeed quite extreme as it is 4 standard deviations away from the most likely value of $8.9 \pm 0.4$ days, computed by us based on the numbers provided by Rodell et al. (2015) and Trenberth et al. (2011).

*These calculations provide a valuable estimate of how long it takes until the global total column water has been depleted to 1/e by precipitation. But can we interpret this measure as a quantitative proxy for the actual moisture residence time, defined as the time water molecules spend in the atmosphere between evaporation and precipitation? Which assumptions go into*

*considering global precipitation as a Poission process? We present here three arguments against such simple 'back-of-the-*

*envelope' calculations. The first and second are related to the assumptions made when following the arguments of the simple estimate. The third one demonstrates that for systems where the assumptions of a Poission process are violated, depletion time constants do not allow to conclude on the moisture residence time.*

All three arguments are falsified below. Moreover, it should be noted that whether or not evaporation and precipitation are Poisson-processes or not this would not affect the global average residence time of water in the atmosphere.

*a. Precipitation is generated in weather systems of different kind and lifetime. Weather systems are formed, may move through the atmosphere, leading to an unequal distribution of precipitation in space and time, until they decay. This 'intermittent' nature of precipitation is a central aspect, and is also related to the atmospheric residence time of water vapor. Some areas of the world experience frequent and heavy precipitation, other areas, such as deserts, hardly experience any rain. Throughout one year, some areas will thus participate more strongly in the atmospheric water cycle than others. This obvious fact becomes important for the simple estimate when considering that the global mean precipitation of 1.69 mm day assumes that all areas of the Earth receive an equal amount of precipitation. According to ERA-Interim reanalyses, 100 % of the global precipitation in a year fall onto 95 % of the global surface area, whereas 96 % fall onto less than 80 % of the surface area (Fig. C1). If we redo the simple estimate from above taking this fact into account, we have to correct the global average rain rates for the actual surface area participating in the water cycle. This would lead us to conclude for example that 90 % of the effective global precipitation from the simple number example (1.37/0.8=1.71 mm day and 1.69/0.8=2.11 mm day, respectively) give depletion time constants of 7.2 and 5.8 days, respectively. While somewhat large values may have been selected here, this sensitivity points out that by considering the world's arid areas appropriately results in shorter residence times, in fact already quite a bit closer to the about 4-5 days we obtain from the LS16 method. The simple estimates rely thus on a global uniform distribution of precipitation, and global participation of all surface area, which is in fact not given. In terms of a Poission process, the spatial and temporal coherence of precipitation violates the randomness requirement.*

Again, Sodemann is mixing up units here. Global mean precipitation for the numbers used in the example by Sodemann should be $616 \cdot 10^3 \, \mathrm{km}^3 \, \mathrm{yr}^{-1} = 1.69 \cdot 10^3 \, \mathrm{km}^3 \, \mathrm{day}^{-1} = 3.31 \, \mathrm{mm} \, \mathrm{day}^{-1}$ for the extreme case and $500 \cdot 10^3 \, \mathrm{km}^3 \, \mathrm{yr}^{-1} = 1.37 \cdot 10^3 \, \mathrm{km}^3 \, \mathrm{day}^{-1} = 2.68 \, \mathrm{mm} \, \mathrm{day}^{-1}$ for the normal case.

To "correct" the global average value for the 80 % of the Earth's surface that receives most precipitation has no point when talking about the global average value. Recall that in VT16 Eqs (2) and (3) no use has been made of the surface area of the Earth. Intuitively, a global average value concerns a precipitation-weighted value or bulk value. When trying to "correct" for the wettest regions of the Earth only, one essentially calculates the spatial average residence time of precipitation for of the x % wettest regions of the Earth, which is obviously not the same as the global average residence time of water in the atmosphere. In LS16 it is nowhere mentioned that they understand their estimate of 4–5 days as a spatial average precipitation residence time for the 80 % wettest regions, but instead present the 4–5 days estimate as the global mean atmospheric moisture residence time.

The mere fact that precipitation varies in space and time has nothing to do with the global mean value, nor does a bulk estimation invoke the necessity of being a Poisson process. In our manuscript (VT16) we clearly showed the spatial and

[Figure]

**Figure C 1.** Surface area fraction vs. precipitation fraction from the ERA-Interim reanalysis data (red line). The dashed line would result if precipitation were spatially homogeneous. Reproduced from Läderach and Sodemann (2016).

temporal variability of the residence time as computed by our tracking methods and still we arrive at global average residence times of 8–10 days.

As a side-note, the numeric examples provided by Sodemann are slightly different than those provided in LS16 (Supplement Section 4). In both cases, however, the spatial average calculation is not correctly executed: the units are mixed-up and the calculation itself is wrong. As shown above, the value 1.37 should have had units $10^3 \, \mathrm{km}^3 \, \mathrm{day}^{-1}$, of which 96 % falls over 80 % of the land surface. Thus, $1.37 * 0.96 = 1.32 \cdot 10^3 \, \mathrm{km}^3 \, \mathrm{day}^{-1}$. However, to be able to calculate a spatial average one should also know how much atmospheric storage (in $10^3 \, \mathrm{km}^3$) resides over that 80 % land surface. Dividing that number by $1.32 \cdot 10^3 \, \mathrm{km}^3 \, \mathrm{day}^{-1}$ will give the spatial average residence time in days over 80 % of the land surface. We did not compute this here as it is actually irrelevant in the discussion as we have not addressed the spatial average in our manuscript anyway.

*b. Due to the prevalent atmospheric stratification, different time scales may be relevant at different atmospheric layers. For example, a lower layer of the atmosphere, representing 50 % of the column water, or integrated water vapour (IWV), may precipitate and recharge much faster with shallow weather systems. Tropical deep convection may involving the entire column water into precipitation generation, but only exist in some regions such as the ITCZ. A residence time would always consider rain to originate from the entire column, thus neclecting the existence of a faster branch. The assumption that all moisture would be depleted by a "deep" process may contribute to overly large estimates of the moisture residence time from depletion time*

[Figure]

**Figure C 2.** Examples of different probability distributions of global atmospheric residence time, which all have a mean of 8.4 days. The "true" distribution is not exactly known, but this figure serves to illustrate that different mixing assumptions may skew the distribution without affecting the global mean. a) Probability density functions. b) Cumulative distribution functions.

*constants. One may consider that a combination of several Poission processes could represent this complexity in a statistical framework.*

Indeed there may very well be water particles that recycle much faster than the average particles. On the other hand, there will also be water particles that recycle much slower in the case they are found in atmospheric layers above the precipitation processes. However, wind speeds above the atmospheric boundary layer are generally quite high and these water particles will be transported, sooner or later, to places where there is deep convection or other processes that cause vertical mixing. Thus, also these water particles participate in the hydrological cycle. The argument of stratification only holds if there are water particles that never participate in precipitation processes. Even if we make the – perhaps not totally strange assumption – that water outside the troposphere (thus in the stratosphere or mesosphere) does not participate in the hydrological cycle (i.e., dead storage), the atmospheric storage is reduced by only $\approx 1\,\%$. The corresponding global average residence time of water in the troposphere then is $8.8\pm0.4\,\mathrm{days}$ (using the numbers from VT16, Fig 1). As we think it is important to be aware of these numbers we will add a sentence about this in Section 3 of the revised manuscript.

In both WAM-2layers and 3D-Trajectories we assume evaporation to enter in the lower levels of the atmosphere, but for precipitation we have indeed assumed a well-mixed atmosphere. This may have some consequences for the probability density functions (VT16, Fig 4), but can by definition not change the global average. An example of the consequence of the well-mixed assumption for precipitation can be explained at the hand of Fig. C2. Let's assume that the true PDF of global atmospheric residence corresponds to the Gamma($k = 0.8, \theta = 10.5$). Invoking the well-mixed assumption could shift this distribution to become an Exponential($\lambda = 1/8.4$) distribution or Gamma($k = 1.2, \theta = 7$) distribution, but cannot alter the actual mean of the distribution. When more water particles undergo a faster cycle, as a logical consequence, also more water particles undergo a slower cycle.

*c. Consider two hypothetical cases of global temporal precipitation patterns. In the first case, during any given month, rain falls globally every day with an average rain rate of $1.37\,\mathrm{mm\,day^{-1}}$. The same amount of evaporation occurs continously and maintains an atmospheric water volume of $12.7x10^3 km^3$. This case will give a depletion time constant of 9.3 days. In a second case, all of the monthly evaporation happens on the first day of each month, and all of the monthly precipitation on the last day. In this second example, the average lifetime of the water vapour is obviously enhanced considerably, while the depletion time constant would still provide the same value of 9.3 days. Obviously, here the stationarity required by a Possion process is not given. Compared to the real atmosphere, both examples are artificial, but they serve to illustrate the point that the depletion time constants do not necessarily faithfully quantify the residence time of atmospheric water vapour. (This example has been modified from the one given in the Supplement to LS16 to be more to the point of the assumptions underpinning a Possion process).*

Before we show that these examples are inconsistent with themselves and that the "depletion time constant", in fact, equals residence time in both cases, we have copied below the two other hypothetical cases from LS16:

*Consider two hypothetical cases of global temporal precipitation patterns. In the first case, rain falls globally every other day with an efficiency of 100 %, i.e. all atmospheric water vapor rains out. In the second case, it rains once in 30 days, again with an efficiency of 100 %. Evaporation recharges the atmospheric moisture reservoir between the precipitation events in both*

[Figure]

**Figure C 3.** Visualization of four hypothetical cases of depleting and replenishing the global atmospheric water store as described by Sodemann's comment (cases 1 and 2) and LS16 (cases 3 and 4). Note that in panel (a) precipitation and evaporation are constant, and, therefore overlap each other in the visualization.

*cases. Both of these scenarios are not inconsistent with a global long-term average rain rate of $1.37\,\mathrm{mm/day}$ and a global amount of moisture of 12.7 thousands of $\mathrm{km}^3$, and will give a depletion time constant of $9.3\,\mathrm{days}$. Yet the actual residence time of the water vapor in the atmosphere, i.e. the time water vapor stays in the atmosphere between evaporation and precipitation, will be 1 day in the first example, and 30 days in the second. Of course both examples are artificial and unrealistic, but they serve to illustrate the point that the temporal characteristics of global precipitation are not measured by depletion time constants as provided by simple global estimates.*

In the following lines we will explore these four hypothetical cases in more depth, from which we conclude that these four cases, in fact, have very different evaporation/precipitation rates and consequently very different residence times. First, we have again to correct the units that were mixed-up by Sodemann. A global average rain rate of $500 \cdot 10^3\,\mathrm{km}^3\,\mathrm{year}^{-1} = 1.37\,\cdot$

**Table C 1.** Summary of the four hypothetical cases brought forward by Sodemann's comment and LS16.

| Case | Cumulative $P$ or $E$ after 30 days $(10^3\,\mathrm{km}^3)$ | average $P$ or $E$ rate $(10^3\,\mathrm{km}^3\,\mathrm{day}^{-1})$ | Average atmospheric storage $(10^3\,\mathrm{km}^3)$ | Residence time from average age of water during precipitation (days) | Residence time from stock divided by flux (depletion time constant in LS16 terminology) (days) |
|---|---|---|---|---|---|
| 1 | 41.1 | 1.37 | 12.7 | 9.3 | 9.3 |
| 2 | 12.7 | 0.42 | 12.3 | 29.0 | 29.0 |
| 3 | 190.5 | 6.35 | 6.35 | 1.0 | 1.0 |
| 4 | 12.7 | 0.42 | 6.35 | 15.0 | 15.0 |

$10^3\,\mathrm{km}^3\,\mathrm{day}^{-1} = 2.68\,\mathrm{mm}\,\mathrm{day}^{-1}$. The corresponding global average residence time (or depletion time constant as Sodemann calls it) equals $12.7 \cdot 10^3\,\mathrm{km}^3 / 1.37 \cdot 10^3\,\mathrm{km}^3\,\mathrm{day}^{-1} = 9.3\,\mathrm{days}$.

All four cases are displayed graphically in Fig. C3. The corresponding average atmospheric storage, average precipitation rate, average evaporation rate and average residence time are given in Table C1. According to the comment by Sodemann, the
residence time in case 2 should be much greater than the residence time in case 1, but case 1 and 2 would give the exact same depletion time constant. However, we can observe from Fig. C3 and Table C1 that the depletion time constant in case 2 is exactly equal to the residence time based on atmospheric water age. The simple reason for this being that the average precipitation rate in case 2 is much lower than in case 1. According to the Supplement of LS16, case 3 and 4 are both not inconsistent with a long-term average rain rate of $1.37 \cdot 10^3\,\mathrm{km}^3\,\mathrm{day}^{-1}$ and global average atmospheric water storage of $12.7 \cdot 10^3\,\mathrm{km}^3$. As can be
seen from Fig. C3 and Table C1 these cases correspond with neither and the global average precipitation rate in case 1 is many times larger than the precipitation rate in case 2, hence the very different residence times.

According to Sodemann these hypothetical cases were supposed to "demonstrate that for systems where the assumptions of a Poission process are violated, depletion time constants do not allow to conclude on the moisture residence time". By exploring these cases in depth we have clearly shown that residence time can be accurately calculated by dividing a stock by its flux
(compare last two columns of Table C1). In our opinion this is actually quite basic knowledge and we feel that for most readers Section 3 of our manuscript will be clear already as it is. Therefore, we intend to add the exploration of these four cases in a Supplement only, rather than adding it to the paper itself.

**3. The lake analogy**

*In the introduction to their Section 3 (pg. 5, L. 24-27), the authors explain some of the reasoning behind a depletion time constant approach:*

*"Moreover, a lake may be permanently stratified (i.e. there is permanent dead storage) and one could argue that the actual volume participating in the water cycle of the lake does not equal the lake's total volume, meaning that the actual average*

*residence time becomes lower. If one can, however, reliably estimate a lake's volume and in- or outflow, it is not necessary for a lake to be well-mixed for Eq. (2) to hold, the mere necessity is that the entire volume participates in the water cycle. Of course, one could still have significant local differences, but the average can reliably be calculated by Eq. (2)."*

*The lake analogy serves to illustrate some of the main problems when considering the atmosphere as a hydrological reservoir.*
*For a lake, it may be safe to assume some kind of well-mixed behaviour (or participation) on long time scales. Water vapour is however not well-mixed throughout the atmosphere, most water vapour resides close to the surface, and it travels horizontally over limited distances because of precipitation processes. For the lake, water is the medium, in the atmosphere, air is the medium and water is a trace substance. Following the lake analogy strictly, one should rather compare water vapour in the atmosphere to a tracer that is dissolved in the lake water and has source and removal processes at the surface.*

We have nowhere in our manuscript assumed that the entire Earth's atmosphere is well-mixed, we only write that all water in the atmosphere participates in the hydrological cycle. As well as for lakes or the Earth's atmosphere it is completely logical to have spatial variations, but the average can still be calculated by Eq. (2).

*In terms of a Poission process, it may simply be the case that a single random Poisson process does not represent global*
*precipitation adequately. Maybe if one were to use a more realistic representation using several combined Poission processes, or a non-homogeneous Poission process, it may be feasible to obtain a realistic residence time estimate from depletion time constants. While it could be interesting to attempt to represent the atmosphere by a more complex statistical process, we argue that our Lagrangian approach already takes the complexity of the atmosphere into account more realistically that other current approaches.*

We repeat that our methods do not assume Poisson processes, nor constant fluxes. As the spatial patterns observed by LS16 and VT16 are not very different, the transport in the method used by LS16 is comparable in realism to the methods used in VT16. The difference being that the absolute results of LS16 show physically impossible values.

*There are different reservoirs or 'lakes', so to speak, in the atmosphere, some close to the surface that are continuously*
*depleted and replenished by weather systems, and several higher above that only occasionally participate in the atmospheric water cycle, for example during deep convection. The situation varies with latitude and season. If one considers total column water, such as for the global residence time estimate, and as used in the methods of VT16, one implicitly assumes that precipitation extracts water vapour from all atmospheric layers, an assumption that induces large uncertainties in many regions of the world that are dominated by shallower precipitation processes. One consequence of the non-well mixed state of the atmosphere*
*is that one should effectively reduce the IWV in the global average calculation, lowering the residence time. Remaining in the thought framework of a Poission process: If on average 80 % of the column water contribute to precipitation processes, IWV would be again be multiplied by a factor of 0.8, resulting in a 1.5–2 day lower residence time in the two examples above, and thus closely approaching the numbers of the Lagrangian residence time estimate of LS16.*

The statement by Sodemann that if on average 80 % of the column water contributes to precipitation processes yield 1.5–
2 days lower residence times is incorrect. You cannot simply neglect 20 % of the water when that on average does not rain out. When that water does not rain out during a particular storm it continues to "age" and sooner or later that water finds itself travelling vertically due to convection or orography, is caught in a katabatic wind, mixes turbulently for whatever reason, or simply finds itself being part of another precipitation event where 90–100 %, of the column participates. The only possible consequence of the well-mixed assumption for precipitation is that you find somewhat different distributions (see Fig. C2), but it can by definition not affect the average.

*A possibly important issue is the question whether some moisture at high elevation resides in the troposphere for very long times, even months. Of course there is very little total water at these elevations, and one can ask the question whether that moisture should be considered for a residence time estimate if it does not actively take part in the atmospheric water cycle? A*

*meaningful working definition of the residence time of water vapour in the atmosphere could then be more specifically identified as "water vapour in the troposphere that participates in the hydrological cycle on a monthly time scale". Very long-lived water vapour may require different methodological approaches that do not suffer from the accumulation of numerical errors with time.*

  The definition used in our manuscript (VT16) is the residence time of all water in the atmosphere. If Sodemann is more inter- ested in the rather vague definition of "water vapor in the troposphere that participates in the hydrological cycle on a monthly time scale", then he should have clearly stated this in LS16, but this is not the definition we would like to use. Nonetheless, our Fig. 4 shows that these definitions are not that far apart from each other.

**4. Simple estimate of the moisture transport distance**

*Looking at the problem from another direction, one can ask the question, what are the physical consequences of a moisture residence time of 8–10 days? Fig. C4 depicts the global mean humidity-weighted wind velocity over the entire atmospheric column for September 2005 from ERA-Interim reanalyses. Humidity-weighted wind speeds emphasize lower regions of the atmosphere, where most humidity resides, and gives an indication at what speed most of the humidity in the atmospheric column moves during that month. Values are 10–20 $\mathrm{m\,s^{-1}}$ in the mid-latitudes, lower in the subtropics (4–8 $\mathrm{m\,s^{-1}}$) and 8–12 $\mathrm{m\,s^{-1}}$*

*at high latitudes. The implication of this is that moisture would travel on average more than 8000 km before precipitating in the extratropics, and more than 4000 km in the subtropics and tropics. Since the 8–10 days is an average value, individual cases will have substantially longer transport distances associated. Considering for example that mid-latitude weather systems develop and intensify, and thereby readily condense large amounts of water vapour along their fronts during 2–3 days, it is difficult to conceive how that corresponds to a 8–10 day time scale and 8000 km length scale of the water transport. In the*

*subtropics, the distance between the evaporation maxima and the ITCZ is only some 15–20 deg in latitude, and also there it is difficult to understand how the moisture can travel for 4000 km on average before precipitating. Values closer to one half of these would be more consistent with expectations from the weather system characteristics in the respective latitudes. While this is not a proof for a shorter residence time, this argument points out that the 8–10 days are not easily explained, even in light of equally simple metrics of moisture lifetime and transport in the atmosphere as the global depletion time estimate.*

[Figure]

**Figure C 4.** Humidity weighted horizontal wind velocity (for the entire column, layer by layer) during September 2005 from ERA-Interim reanalyses. Unit is $\mathrm{m\,s^{-1}}$. Range rings around the equator indicate distances of 2000 to 8000 km from the point N0 W0. Reproduced from Sodemann's comment

These transport distances are actually quite reasonable and fit very well with the findings of global moisture recycling/transport and isotope studies (e.g., Bosilovich and Schubert 2002; Dirmeyer et al., 2009, 2014; Goessling and Reick 2011; Risi et al., 2013; van der Ent et al., 2014; Yoshimura et al., 2003, 2004a, 2004b). What Sodemann seems to be overlooking is that atmospheric water is most of the time not part of an active storm and may travel quite far during that time. Most particles will not, however, and therefore we also find a median of around 5 days (see VT16, Fig. 4). You can also turn this question around and ask what the consequence is of a 4–5 day residence time? The answer is unrealistically high precipitation rates or unrealistically low atmospheric water storage (see Fig. C3 and Table C1).

**5. There are further important points in VT16 that would merit further explanation or discussion:**

*a. The authors consider 3 variants of the residence time of moisture, termed residence time of precipitation, residence time of evaporation, and age of water vapour. No explanation is given on how the age has been calculated. I assume all of these are different projections of the same quantity (precipitation RT projects forward, evaporation RT projects backward), and should have the same mean value. The relation and difference between each way of presenting the residence time could be stated more clearly to avoid confusion of the readers.*

As mentioned in our response to Jiangfeng Wei (AC3), we will use a shortened version of the Portugal example given in our response to Jiangfeng Wei to clarify the differences between the three metrics displayed in VT16 Figs. 2c-e. The calculation of the age is quite difficult with a Lagrangian method, but very easy with an Eulerian method, thus has been performed for WAM-2layers, and the formula is given in VT16 Eq. (1).

*b. The Tuinenburg method is based on the Dirmeyer and Brubaker (1999) approach (a corresponding reference is missing in the manuscript). As I understand that method, several isentropic trajectories are calculated from every 0.5x0.5deg grid point at several elevations, then surface evaporation is accumulated along these trajectories at every time step. Essentially, that method thereby assumes a well-mixed atmosphere at every time step and grid point - because the vertical position of the trajectory does not matter. Moreover, water vapour is assumed to be a conserved quantity once it is mixed into the air parcel (i.e. precipitation does not remove earlier moisture contributions). The method is furthermore sensitive to the reliability of the evaporation data set. This method clearly relies on strong assumptions, in particular compared to our Lagrangian method (Sodemann et al., 2008) which was applied in LS16 and neither assumes well-mixed conditions nor relies on evaporation, which is a difficult variable to observe and has large local uncertainties, in particular when derived from satellite observations (Rodell et al., 2015).*

Indeed, we should have given reference to Dirmeyer and Brubaker (1999), and we will do so in the revised version of our manuscript. The consequence of the well-mixed assumption for precipitation was already discussed under Section 2b of this response. There is definitely uncertainty in evaporation, and, therefore, we have checked how ERA-Interim evaporation compares to the state-of-the-art estimates from Rodell et al., (2015). This falls within the uncertainty ranges (VT16, Fig. 1). Not making use of evaporation data at all comes at the risk of highly overestimating it (Stohl and James, 2004), and should in our opinion, therefore, not be preferred.

*Interestingly however, the median of the residence time with VT16's Lagrangian method 3D-T are with 5.7 and 4.6 days clearly lower than 8–10 days. There is a lot of very short-lived water vapour identified by this method. VT16 state that the very long tail leads to a mean to 8–10 days. Taken at face value, the low median argues for a residence time of the bulk of the water vapour of much less than 8–10 days. With trajectory calculation times exceeding 10–15 days, the tail gets more and more uncertain, in particular if few trajectories per grid point are considered. What would be the residence time if evaluation was not cut off at 30 (pg. 5 L. 10), but at 20 or, say, 50 days? Would it still be possible to argue that the mean is representative of the distribution? One consequence of this difference between the mean and the median is that the results in Figure 2 of VT16 should be shown separately for the WAM and the 3D-T method.*

The fact that the median is about 5 days has nothing to do with the mean, but only with the distribution. The consequence of 20-day trajectories can easily be read from the PDFs (VT16, Fig. 4). The consequence of 50-day trajectories is not likely to influence the results much as 30-day trajectories already accounted for 95 % of the initial moisture. The location of an individual parcel may be uncertain of 30 days, but all parcels together yield a very smooth shape of the tail of the PDFs. We assumed a 30 day residence for the remaining 5 % of the parcels, and what we can say with certainty is that the mean can impossibly become lower by other assumptions about the shape of the tail.

Figure 2 is already shown separately in the Supplement (Fig. S1).

*c. A particularly puzzling result is shown in Fig. 3 of VT16. The patterns of the seasonal residence time appear difficult to interpret physically. Residence times increase from about 5 days to more than 15 days during northern hemisphere winter*

*over the North Pacific, and the reverse applies to the southern hemisphere. In the current version, VT16 do not provide further explanation on what could cause this result, and how it relates to observed seasonal changes in the climate system. It would be interesting to learn about a corresponding strong change in the climate system that would explain such a drastic seasonal change.*

See our response to Kevin Trenberth AC1 P5:L29–P6:L9.

**References**

Bosilovich, M. G. and Schubert, S. D.: Water vapor tracers as diagnostics of the regional hydrologic cycle, J. Hydrometeorol., 3(2), 149–165, doi:10.1175/1525-7541(2002)003<0149:WVTADO>2.0.CO;2, 2002.

Bosilovich, M. G., Sud, Y., Schubert, S. D. and Walker, G. K.: GEWEX CSE sources of precipitation using GCM water vapor tracers, GEWEX News, 12(3), 1,6–7,12, 2002.

Chow, V. T., Maidment, D. R. and Mays, L. W.: Applied Hydrology, McGraw-Hill, Singapore., 1988.

Dee, D. P., Uppala, S. M., Simmons, A. J., Berrisford, P., Poli, P., Kobayashi, S., Andrae, U., Balmaseda, M. A., Balsamo, G., Bauer, P., Bechtold, P., Beljaars, A. C. M., van de Berg, L., Bidlot, J., Bormann, N., Delsol, C., Dragani, R., Fuentes, M.,

Geer, A. J., Haimberger, L., Healy, S. B., Hersbach, H., Hólm, E. V, Isaksen, L., Kållberg, P., Köhler, M., Matricardi, M., McNally, A. P., Monge-Sanz, B. M., Morcrette, J. J., Park, B. K., Peubey, C., de Rosnay, P., Tavolato, C., Thépaut, J. N. and Vitart, F.: The ERA-Interim reanalysis: Configuration and performance of the data assimilation system, Q. J. R. Meteorol. Soc., 137(656), 553–597, doi:10.1002/qj.828, 2011.

Dirmeyer, P. A. and Brubaker, K. L.: Contrasting evaporative moisture sources during the drought of 1988 and the flood of

1993, J. Geophys. Res., 104(D16), 19383–19397, doi:10.1029/1999JD900222, 1999.

Dirmeyer, P. A., Brubaker, K. L. and DelSole, T.: Import and export of atmospheric water vapor between nations, J. Hydrol., 365(1–2), 11–22, doi:10.1016/j.jhydrol.2008.11.016, 2009.

Fitzmaurice, J. A.: A critical Analysis of Bulk Precipitation Recycling Models, Thesis, Massachusetts Institute of Technology, Massachusetts., 2007.

Goessling, H. F. and Reick, C. H.: What do moisture recycling estimates tell us? Exploring the extreme case of non-evaporating continents, Hydrol. Earth Syst. Sci., 15(10), 3217–3235, doi:10.5194/hess-15-3217-2011, 2011.

Goessling, H. F. and Reick, C. H.: On the "well-mixed" assumption and numerical 2-D tracing of atmospheric moisture, Atmos. Chem. Phys., 13(11), 5567–5585, doi:10.5194/acp-13-5567-2013, 2013.

Hendriks, M. R.: Introduction to Physical Hydrology, Oxford University Press, New York., 2010.

Jones, J. A. A.: Global Hydrology: processes, resources and environmental management, Longman, Harlow., 1997.

Läderach, A. and Sodemann, H.: A revised picture of the atmospheric moisture residence time, Geophys. Res. Lett., 43, 924–933, doi:10.1002/2015GL067449, 2016.

Monsen, N. E., Cloern, J. E., Lucas, L. V. and Monismith, S. G.: A comment on the use of flushing time, residence time, and age as transport time scales, Limnol. Oceanogr., 47(5), 1545–1553, doi:10.4319/lo.2002.47.5.1545, 2002.

Rodell, M., Beaudoing, H. K., L'Ecuyer, T. S., Olson, W. S., Famiglietti, J. S., Houser, P. R., Adler, R., Bosilovich, M. G., Clayson, C. A., Chambers, D., Clark, E., Fetzer, E. J., Gao, X., Gu, G., Hilburn, K., Huffman, G. J., Lettenmaier, D. P., Liu, W. T., Robertson, F. R., Schlosser, C. A., Sheffield, J. and Wood, E. F.: The observed state of the water cycle in the early twenty-first century, J. Clim., 28(21), 8289–8318, doi:10.1175/JCLI-D-14-00555.1, 2015.

Savenije, H. H. G.: Water scarcity indicators; the deception of the numbers, Phys. Chem. Earth Part B Hydrol. Ocean. Atmos., 25(3), 199–204, doi:10.1016/S1464-1909(00)00004-6, 2000.

Schicker, I., Radanovics, S. and Seibert, P.: Origin and transport of Mediterranean moisture and air, Atmos. Chem. Phys., 10(11), 5089–5105, doi:10.5194/acp-10-5089-2010, 2010.

Sodemann, H., Schwierz, C. and Wernli, H.: Interannual variability of Greenland winter precipitation sources: Lagrangian
moisture diagnostic and North Atlantic Oscillation influence, J. Geophys. Res. D Atmos., 113(3), D03107, doi:10.1029/2007jd008503, 2008.

Stohl, A. and James, P.: A Lagrangian analysis of the atmospheric branch of the global water cycle: Part 1: Method description, validation, and demonstration for the August 2002 flooding in central Europe, J. Hydrometeorol., 5(4), 656–678, 2004.

Stohl, A. and Seibert, P.: Accuracy of trajectories as determined from the conservation of meteorological tracers, Q. J. R.
Meteorol. Soc., 124(549), 1465–1484, doi:10.1256/smsqj.54906, 1998.

Trenberth, K. E.: Atmospheric moisture residence times and cycling: Implications for rainfall rates and climate change, Clim. Change, 39(4), 667–694, doi:10.1023/A:1005319109110, 1998.

Trenberth, K. E., Fasullo, J. T. and Mackaro, J.: Atmospheric Moisture Transports from Ocean to Land and Global Energy Flows in Reanalyses, J. Clim., 24(18), 4907–4924, doi:10.1175/2011jcli4171.1, 2011.

UCAR: The Water Cycle, [online] Available from: http://scied.ucar.edu/longcontent/water-cycle (Accessed 22 July 2016), 2011.

van der Ent, R. J. and Savenije, H. H. G.: Length and time scales of atmospheric moisture recycling, Atmos. Chem. Phys., 11(5), 1853–1863, doi:10.5194/acp-11-1853-2011, 2011.

van der Ent, R. J. and Tuinenburg, O. A.: The residence time of water in the atmosphere revisited, Hydrol. Earth Syst. Sci.
Discuss., in review, doi:10.5194/hess-2016-431, 2016.

van der Ent, R. J., Wang-Erlandsson, L., Keys, P. W. and Savenije, H. H. G.: Contrasting roles of interception and transpiration in the hydrological cycle - Part 2: Moisture recycling, Earth Syst. Dyn., 5(2), 471–489, doi:10.5194/esd-5-471-2014, 2014.

Ward, R. C. and Robinson, M.: Principles of Hydrology, McGraw-Hill, Berkshire., 2000.

Yoshimura, K., Oki, T., Ohte, N. and Kanae, S.: A quantitative analysis of short-term O-18 variability with a Rayleigh-type
isotope circulation model, J. Geophys. Res., 108(D20), doi:10.1029/2003jd003477, 2003.

Yoshimura, K., Oki, T., Ohte, N. and Kanae, S.: Colored moisture analysis estimates of variations in 1998 Asian monsoon water sources, J. Meteorol. Soc. Japan, 82(5), 1315–1329, doi:10.2151/jmsj.2004.1315, 2004a

Yoshimura, K., Oki, T. and Ichiyanagi, K.: Evaluation of two-dimensional atmospheric water circulation fields in reanalyses by using precipitation isotopes databases, J. Geophys. Res., 109(D20), doi:10.1029/2004jd004764, 2004b.

---

## Author Comment (AC7) · 15 Nov 2016

We would like to thank everyone that has participated in this interactive discussion. Based on the reviews we have received from Kevin Trenberth, Jiangfeng Wei, Anonymous Referee #3 and Harald Sodemann, we intend to incorporate a number of changes into the manuscript (van der Ent and Tuinenburg (2016). Below, we provide a bullet-wise summary:

- Add a non-exhaustive table to the introduction, listing the residence times (or other time scales) found in previous studies;

- More explicit statement of assumptions for the atmospheric moisture tracking models in Section 2.2;

- Insert a more elaborative summary of the main points brought forward by Läder-ach and Sodemann (2016) against the use of Eqs. (2) and (3). And, of course, a summary of our counterarguments against it;

- Remove the speculative statement in Section 3 about irregularities in the method of Läderach and Sodemann (2016). Instead replace this with a more factual statement in the introduction regarding the assumptions made in that method and how that may bias their results towards lower residence times;

- Discuss what would be the result of assuming that water outside the troposphere does not participate in the hydrological cycle (Section 3);

- A more clear and in-depth explanation (Portugal example) of the differences between precipitation residence time, evaporation residence time and age of atmospheric water in Section 4;

- A more in-depth discussion of the difference between atmospheric moisture age in January and July;

- Discuss in Section 5 how the assumptions in the method and how ERA-Interim data could influence these results;

- Discussion of the differences in results for WAM-2layers and 3D-T in the Supplement;

- Add a section to the Supplement that extensively investigates, discusses and falsifies the arguments brought forward against the use of Eqs. (2) and (3), by Läderach and Sodemann (2016) and the comment of Sodemann SC1;

- Add a figure to the Supplement that gives atmospheric storage, precipitation and evaporation for January and July to help the interpretation of the seasonal variations in Fig. 3 and Fig. S2.

**References**

Läderach, A. and Sodemann, H.: A revised picture of the atmospheric moisture residence time, Geophys. Res. Lett., 43, 924–933, doi:10.1002/2015GL067449, 2016.

van der Ent, R. J. and Tuinenburg, O. A.: The residence time of water in the atmosphere revisited, Hydrol. Earth Syst. Sci. Discuss., in review, doi:10.5194/hess-2016-431, 2016.

---

## Author Response (AR1)

Below we respond to the editor decision
* * *
*Editor Decision: Reconsider after major revisions (further review by Editor and Referees) (16 Nov 2016) by Hannah Cloke*

*Comments to the Author:*

*This manuscript generated a lively scientific discussion. As the authors have pointed out, the discussion is now much longer than the manuscript itself. This is clearly an important and timely topic for the HESS readership.*

*Overall I find that although there is scientific merit in this work, the conclusions are too strong and the assumptions not clearly enough defined.*

*In particular:*

*1. I do not agree with the authors that they have "clearly stated our assumptions and presented our results under these assumptions". I do not find the assumptions easy to follow. Many of them are not made explicit and the burden of understanding the assumption is placed on the reader already knowing or reading about it elsewhere. Please can you take more space to identify the consequences of the modelling and parameterizations - i.e. the impacts on and uncertainties in your results and conclusions. Although it is common to say that 'model world' results are applicable to the real world it would be much more robust to present these alongside a clear indication of the main assumptions. I would suggest a whole section on assumptions and their consequences would be one way to address this.*

*2. I also find the conclusion of 'disproving' LS16's estimates not adequately supported by the evidence provided. Your work does suggest a discrepancy but you do not show evidence that disproves this. I would agree with the comments of the Referees and other comments here. You will need to modify your statements. A much clearer explanation of why there might be differences between different methods would be valuable here.*

*3. A much clearer review of LS16 and other atmospheric residence time work would no doubt help the readability of the manuscript and assist the reader in following your arguments. Please take some space to undertake this.*

*Your summary of anticipated changes to your manuscript addresses to some degree most of the main referee comments and my three main concerns made above and thus I look forward to receiving your revised manuscript.*
* * *
Dear editor,

We are grateful for the fact that you see the importance, timeliness and scientific merit of our work. In the revised version we have highlighted our assumptions and the effect on our results more clearly.

We agree that our assumptions could have been stated more clearly. In the revised manuscript, we have, however, not added on whole section of assumptions and consequences, but we have discussed the effect of our assumptions, or assumptions made in other studies, or assumptions in general, alongside the presented results in the manuscript. Moreover, we have added a disclaimer in our conclusions section.

Indeed, we would like to show the reader that the established knowledge of a global average residence time of atmospheric water being 8-10 days must be valid. As a consequence, the estimate of 4-5 days by LS16 cannot be valid, however, to pinpoint exactly why LS16 find such a low residence time would require the set-up of a whole new study, collaborative experiments with the group of Sodemann, and preferably another group with an online tracking method. We would very much like to do this, but this goes beyond the scope of this paper. In the revised manuscript, we have done all we can to show why we think the 8-10 days estimate must be valid. We have approached the question from a physical mass balance point-of-view, give factual statements about assumptions not addressed in LS16, and have now included a supplement that, in our opinion, gives strong counterarguments against the counterarguments by LS16. Following your suggestion, we have removed any strong vocabulary and improved the clarity of our review of previous work. We think that we have built a strong case for the 8-10 days estimate, but in the end it is up to the reader to decide, which line of thought he/she agrees with.

That being said, we actually hope that the reader mainly focuses on the innovative results presented in our figures rather than on the global average estimate discrepancy with LS16. We look forward to a continued review process.

Utrecht, 25-11-2016

Ruud van der Ent
Obbe Tuinenburg

Summary
25-11-2016 10:58:40

Differences exist between documents.

**New Document:**
VanderEnt_and_Tuinenburg2016_final_v2
20 pages (6.99 MB)
25-11-2016 10:58:33
Used to display results.

**Old Document:**
hess-2016-431-manuscript-version2
15 pages (6.95 MB)
25-11-2016 10:58:33

Get started: first change is on page 1.

No pages were deleted

**How to read this report**

Highlight indicates a change.
Deleted indicates deleted content.
▲ indicates pages were changed.
⟷ indicates pages were moved.

[revised manuscript text omitted]

Summary
18-11-2016 15:20:56

Differences exist between documents.

**New Document:**
VanderEnt_and_Tuinenburg2016_supplement_v2
10 pages (11.89 MB)
18-11-2016 15:20:55
Used to display results.

**Old Document:**
hess-2016-431-supplement-version2
2 pages (6.41 MB)
18-11-2016 15:20:55

No pages were deleted

**How to read this report**

**Highlight** indicates a change.
 indicates deleted content.
▲ indicates pages were changed.
⬌ indicates pages were moved.

**The residence time of water in the atmosphere revisited**

Ruud J. van der Ent[1] and Obbe A. Tuinenburg[2]

[1]Department of Physical Geography, Faculty of Geosciences, Utrecht University, Utrecht, the Netherlands
[2]Department of Environmental Sciences, Copernicus Institute for Sustainable development, Utrecht University, Utrecht, the Netherlands

*Correspondence to:* Ruud J. van der Ent (r.j.vanderent@uu.nl)

**Supplement**

**Contents**

**1 Supplementary Figures S1–S3 and short discussion of the differences in results from WAM-2layers and 3D-T**

[Figure]

**Figure S 1.** Annual average atmospheric residence times for 2002–2008, based on ERA-Interim data. **(a)** Residence time of precipitation as computed by WAM-2layers. **(b)** Residence time of precipitation as computed by 3D-Trajectories. **(c)** as (a) for evaporation. **(d)** As (b) for evaporation.

Figure S1 shows the separate estimates from WAM-2layers and 3D-T, for which the merged estimates are shown in Fig. 2c+d
5   of the main paper. Overall, the patterns are very similar, but 3D-T estimates higher residence times of about 1 day on average. Moreover, the patterns for WAM-2layers are much smoother compared to the patterns of 3D-T. We attribute this difference to the fact that WAM2-layers was run on a coarser resolution and has numerical diffusion.

In the Eulerian WAM-2layers model, we find slightly lower residence times than what would be expected from the global water balance. One reason for this is that the tagged water column in a certain grid cell could get 'full' due to the fact that the
10   offline computed water balance by WAM-2layers is not necessarily equal to the ERA-Interim water balance, which is a more detailed model, but also includes data-assimilation. This leads to a well-known unphysical residual (see Dominguez et al., 2006; Bisselink and Dolman, 2008, for a more elaborate discussion). When the tagged water reservoir is 'full' we let this water disappear from the model, however, in the next time step this water may again be mixed with newly evaporated water. Due to the fact that some water disappeared in the previous time step, the resulting age may be biased towards lower values. Something similar happens over the northern and southern boundaries (at 80° latitudes), where water disappears over the boundaries.

In the Lagrangian 3D-T model, however, we find slightly higher residence times than what would be expected from the global water balance. The mixing in 3D-T is based on 6-hourly vertical wind speeds, but in reality there might be more turbulent vertical mixing. It is, therefore, possible that the water in 3D-T remains in the upper atmosphere for too long leading to slightly higher residence times. Moreover, in 3D-T, we have performed trajectory calculations until 90° latitude and we set back the trajectories if they 'disappear' over these polar boundaries, possibly leading to overestimated residence times.

All in all, we think that the merged estimate (Fig. 2 in the main paper) gives the values that are closest to the 'true' residence time. For reason given above, WAM2-layers can be biased to lower values, whereas 3D-T can be biased to higher values. The merged estimate compensates for assumptions made in both models.

[Figure]

**Figure S 2.** Atmospheric residence times in January and July for 2002–2008, based on ERA-Interim data (averages of WAM-2layers and 3D-T).

[Figure]

**Figure S 3.** Atmospheric moisture storage (i.e., precipitable water), precipitation and evaporation in January and July (2002—2008, ERA-Interim).

**2 Discussion of the disputed validity of global water balance calculations for the global average residence time of water in the atmosphere**

5 As discussed in our manuscript, the commonly accepted method to calculate the average residence time of any reservoir is to divide its mass (or volume) by the influx or outflux under the assumption that the total mass does not change over a longer time period. This was, however, challenged by Läderach and Sodemann (2016) and they provided two counter-arguments against such simple calculations in section 4 of their Supporting Information. These two arguments are iterated below, and we will argue that these counter-arguments are not valid. First, we have to introduce the global numbers, which are used in the examples

10 by (Läderach and Sodemann, 2016). We quote these numbers below, but we took the liberty to directly 'correct' the mistakes that were made in the units. Please refer to author comment 6 (AC6) or short comment 1 (SC1) in the interactive discussion for the 'uncorrected' quotes.

Läderach and Sodemann (2016) write:

15 *"Using a value for global precipitation (which equals global evaporation) of $500 \cdot 10^3 \, \text{km}^3/\text{yr} = 1.37 \cdot 10^3 \, \text{km}^3 \, \text{day}^{-1} = 2.68 \, \text{mm} \, \text{day}^{-1}$ and a volume of the global moisture in the atmosphere of $12.7 \cdot 10^3 \, \text{km}^3$ (Trenberth et al., 2011), one obtains a global depletion time constant of $12.7 \cdot 10^3 / 1.37 \cdot 10^3 = 9.3 \, \text{days}$. Assuming a more extreme case within the range of uncertainty for both quantities, the numbers change to a global precipitation of $616 \cdot 10^3 \, \text{km} \, \text{yr}^{-1} = 1.69 \, \text{km}^3 \, \text{day}^{-1} = 3.31 \, \text{mm} \, \text{day}^{-1}$ and the global amount of moisture in the atmosphere of $12.3 \cdot 10^3 \, \text{km}^3$. This would result in a global average depletion time*

20 *constant, assumed to be identical to the residence time of moisture, of $12.3 \cdot 10^3 / 1.69 \cdot 10^3 = 7.3 \, \text{days}$."*

Before we present the two counter-arguments by (Läderach and Sodemann, 2016), it should be noted that the extreme case is indeed quite extreme as it is 4 standard deviations away from the most likely value of $8.9 \pm 0.4 \, \text{days}$, computed by us based on the numbers provided by Rodell et al. (2015) and Trenberth et al. (2011).

The first counter-argument by Läderach and Sodemann (2016) is:

*"Precipitation is generated in weather systems of different kind and lifetime. Weather systems are formed, may move through the atmosphere, leading to an unequal distribution of precipitation in space and time, until they decay. This 'intermittent' nature of precipitation is a central aspect, and is also related to the atmospheric residence time of water vapor. Some areas of the world experience frequent and heavy precipitation, other areas, such as deserts, hardly experience any rain. Throughout one year,*

30 *some areas will thus participate more strongly in the atmospheric water cycle than others. This obvious fact becomes important for the simple example when considering that the global mean precipitation of $1.69 \cdot 10^3 \, \text{km}^3 \, \text{day}^{-1} = 3.31 \, \text{mm} \, \text{day}^{-1}$ assumes that all areas of the earth receive an equal amount of precipitation. According to ERA-Interim, $90\%$ of the global precipitation in a year fall onto less than $70\%$ of the global surface area (Fig. S4). If we redo the simple estimate from above taking this fact into account, we have to correct the global average rain rates for the actual surface area participating in the water cycle. This would lead us to conclude that $90\%$ of the effective global precipitation from the simple number example ($1.37/0.7 =$*

*1.95 km³ day⁻¹ and 1.69/0.7 = 2.41 km³ day⁻¹, respectively) lead us to depletion time constants of 6.51 and 5.10 days, respectively. This is clearly shorter than the 7–9 days obtained originally, and in fact quite close to the about 4–5 days we*
5 *obtain from our method. The simple estimates rely thus on a global uniform distribution of precipitation, which is in fact not given."*

In the interactive discussion (SC1), Sodemann adds: *"In terms of a Poisson process, the spatial and temporal coherence of precipitation violates the randomness requirement."*

[Figure]

**Figure S 4.** Surface area fraction vs. precipitation fraction from the ERA-Interim reanalysis data (red line). The dashed line would result if precipitation were spatially homogeneous. Reproduced from Läderach and Sodemann (2016).

10 We disagree with this argument, because it there is no point in 'correcting' the value for the 70 % of the Earth's surface that receives most precipitation, when talking about the global average value. Recall that in the main paper (Eqs (2) and (3)) no use has been made of the surface area of the Earth. Intuitively, a global average value concerns a precipitation-weighted value or bulk value. When trying to 'correct' for the wettest regions of the Earth only, one essentially calculates the spatial average depletion time of precipitation for of the x % wettest regions of the Earth, which is obviously not the same as the global average

15 residence time of water in the atmosphere. In Läderach and Sodemann (2016) it is nowhere mentioned that they understand their estimate of 4–5 days as a spatial average precipitation residence time for the 70 % wettest regions, but instead present the 4–5 days estimate as the global mean atmospheric moisture residence time.

The mere fact that precipitation varies in space and time has no effect on the global mean value, nor does a bulk estimation invoke the necessity of being uniform, constant or a Poisson process. In the main paper we clearly showed the spatial and

temporal variability of the residence time as computed by our tracking methods and still we arrive at global average residence times of 8–10 days.

5     As a side-note, the spatial average calculation is not correctly executed. The global average precipitation is $1.37 \cdot 10^3\,\mathrm{km}^3\,\mathrm{day}^{-1}$, of which 90 % falls over 70 % of the land surface. Thus, $1.37 * 0.90 = 1.23 \cdot 10^3\,\mathrm{km}^3\,\mathrm{day}^{-1}$. However, to be able to calculate a spatial average one should also know how much atmospheric storage (in $10^3\,\mathrm{km}^3$) resides over that 70 % land surface. Dividing that number by $1.23 \cdot 10^3\,\mathrm{km}^3\,\mathrm{day}^{-1}$ will give the spatial average depletion time in days over 70 % of the land surface. We did not compute this here as it is actually not a pure residence time, and we have not addressed spatial averages in our main paper 10 anyway.

The second counter-argument by Läderach and Sodemann (2016) involves the description of two cases that, in their words, *"criticizes the simple estimate more fundamentally and demonstrates that depletion time constants do not allow to conclude on the moisture residence time."* In the interactive discussion (SC1), Sodemann adds two different cases. Here, we explore these 15 four hypothetical cases in more depth.

Sodemann writes in SC1:

*"Consider two hypothetical cases of global temporal precipitation patterns. In the first case, during any given month, rain falls globally every day with an average rain rate of $1.37 \cdot 10^3\,\mathrm{km}^3\,\mathrm{day}^{-1}$. The same amount of evaporation occurs continuously* 20 *and maintains an atmospheric water volume of $12.7 \cdot 10^3\,\mathrm{km}^3$. This case will give a depletion time constant of $9.3$ days. In a second case, all of the monthly evaporation happens on the first day of each month, and all of the monthly precipitation on the last day. In this second example, the average lifetime of the water vapor is obviously enhanced considerably, while the depletion time constant would still provide the same value of $9.3$ days. Obviously, here the stationarity required by a Poisson process is not given. Compared to the real atmosphere, both examples are artificial, but they serve to illustrate the point that* 25 *the depletion time constants do not necessarily faithfully quantify the residence time of atmospheric water vapor."*

    And Läderach and Sodemann (2016) write: *"Consider two hypothetical cases of global temporal precipitation patterns. In the first case, rain falls globally every other day with an efficiency of 100 %, i.e. all atmospheric water vapor rains out. In the second case, it rains once in 30 days, again with an efficiency of 100 %. Evaporation recharges the atmospheric moisture reservoir between the precipitation events in both cases. Both of these scenarios are not inconsistent with a global long-term* 30 *average rain rate of $1.37 \cdot 10^3\,\mathrm{km}^3\,\mathrm{day}^1$ and a global amount of moisture of $12.7 \cdot 10^3\,\mathrm{km}^3$, and will give a depletion time constant of $9.3$ days. Yet, the actual residence time of the water vapor in the atmosphere, i.e. the time water vapor stays in the atmosphere between evaporation and precipitation, will be $1$ day in the first example, and $30$ days in the second. Of course both examples are artificial and unrealistic, but they serve to illustrate the point that the temporal characteristics of global precipitation are not measured by depletion time constants as provided by simple global estimates.*

35

All four cases are displayed graphically by us in Fig. S5. The corresponding average atmospheric storage, average precipitation rate, average evaporation rate and average residence time are given in Table S1. According to Sodemann (SC1), the residence

[Figure]

**Figure S 5.** Visualization of four hypothetical cases of depleting and replenishing the global atmospheric water store as described by Sodemann's comment (cases 1 and 2) and LS16 (cases 3 and 4). Note that in panel (a) precipitation and evaporation are constant, and, therefore overlap each other in the visualization.

time in case 2 should be much greater than the residence time in case 1, but case 1 and 2 would give the exact same depletion time constant. However, we can observe from Fig. S5 and Table S1 that the depletion time constant in case 2 is exactly equal to the residence time based on atmospheric water age. The simple reason for this being that the average precipitation rate in case 2 is much lower than in case 1. According to the Supplement of Läderach and Sodemann (2016), case 3 and 4 are both *"not inconsistent with a long-term average rain rate of* $1.37 \cdot 10^3 \, \text{km}^3 \, \text{day}^{-1}$ *and global average atmospheric water storage of* $12.7 \cdot 10^3 \, \text{km}^3$*"*. As can be seen from Fig. S5 and Table S1, however, these cases correspond with neither and the global average precipitation rate in case 1 is many times larger than the precipitation rate in case 2, hence the very different residence times.

According to Sodemann (SC1) these hypothetical cases were supposed to *"demonstrate that for systems where the assumptions of a Poisson process are violated, depletion time constants do not allow to conclude on the moisture residence time"*.

**Table 1.** Summary of the four hypothetical cases brought forward by Sodemann (SC1) and Läderach and Sodemann (2016).

| Case | Cumulative $P$ or $E$ after 30 days $(10^3 \, km^3)$ | average $P$ or $E$ rate $(10^3 \, km^3 \, day^{-1})$ | Average atmospheric storage $(10^3 \, km^3)$ | Residence time from average age of water during precipitation (days) | Residence time from stock divided by flux (depletion time constant in Läderach and Sodemann's terminology) (days) |
|---|---|---|---|---|---|
| 1 | 41.1 | 1.37 | 12.7 | 9.3 | 9.3 |
| 2 | 12.7 | 0.42 | 12.3 | 29.0 | 29.0 |
| 3 | 190.5 | 6.35 | 6.35 | 1.0 | 1.0 |
| 4 | 12.7 | 0.42 | 6.35 | 15.0 | 15.0 |

By exploring these cases in depth we have clearly shown that residence time can, in fact, be accurately calculated by dividing a stock by its flux (compare last two columns of Table S1). We conclude that the examples provided by Läderach and Sodemann (2016) and SC1 are inconsistent with themselves and do not show that the 'depletion time constant' is different from the residence time, as, in fact, these equal each other in all four cases (Table S1, last two columns).

---

## Author Response (AR2)

Below we respond to the editor decision and reviewer comments
* * *
*Editor Decision: Publish subject to minor revisions (further review by Editor) (13 Jan 2017) by Hannah Cloke*

*Comments to the Author:*
*The Referees find your manuscript to be of good quality and suggest only one minor point that requires addressing before publication. Please check this and revise your manuscript accordingly.*

Response:
Dear editor, below we respond to the issue raised by one of the reviewers and we have uploaded an updated supplement. We would like to thank you for the handling of the manuscript

*Reviewer Jiangfeng Wei:*
*Thank the authors for addressing all those comments.*
*The section 2 of the Supplement is interesting. It discusses the arguments of Läderach and Sodemann (2016) point-by-point. I have a question on Figure S5 and Table S1. According to my understanding of their description, case 2 should have the same cumulative P or E as case 1, i.e., 41.1 (10^3 km^3). I don't know why you got a value of 12.7 (10^3 km^3). Similar for case 4 relative to case 3.*

Response:
The description of some cases is open to different interpretation due to the fact that they are internally conflicting. In case 2 we cannot both have a globally average precipitation of 1.37 (10^3 km^3 day^-1) and an atmospheric water volume of 12.7 (10^3 km^3). We decided to assume the upper bound of the atmospheric reservoir to be 12.7 (10^3 km^3). However, we could have alternatively assumed the flux of 1.37 (10^3 km^3 day^-1) to be applicable in all cases and derive the storage from that starting point instead. This is similarly true for cases 3 and 4. However, we agree with Jiangfeng Wei that it would have been clearer to use the value of 1.37 (10^3 km^3 day^-1) for all four cases and derive the resulting atmospheric storage over time. We have updated the supplement accordingly, but the conclusions regarding these cases has remained unchanged.